# SIMULTANEOUS REWARD DISTILLATION AND PREFERENCE LEARNING: GET YOU A LANGUAGE MODEL WHO CAN DO BOTH

## ABSTRACT

Reward modeling of human preferences is one of the cornerstones of building usable generative large language models (LLMs). While traditional RLHF-based alignment methods explicitly maximize the expected rewards from a separate reward model, more recent supervised alignment methods like Direct Preference Optimization (DPO) circumvent this phase to avoid problems including model drift and reward overfitting. Although popular due to its simplicity, DPO and similar direct alignment methods can still lead to degenerate policies, and rely heavily on the Bradley-Terry-based preference formulation to model reward differences between pairs of candidate outputs. This formulation is challenged by non-deterministic or noisy preference labels, for example human scoring of two candidate outputs is of low confidence. In this paper, we introduce **DRDO (Direct Reward Distillation and policy-Optimization)**, a supervised knowledge distillation-based preference alignment method that simultaneously models rewards *and* preferences to avoid such degeneracy. DRDO directly mimics rewards assigned by an oracle while learning human preferences from a novel preference likelihood formulation. Our experimental results on the Ultrafeedback and TL;DR datasets demonstrate that policies trained using DRDO surpass previous methods such as DPO and e-DPO in terms of expected rewards and are more robust, on average, to noisy preference signals as well as out-of-distribution (OOD) settings.

## 1 INTRODUCTION

In the development of large language models (LLMs), robust modeling of human preferences is essential for producing outputs that are contextually and situationally appropriate, in order to result in models that users will actually want to use. Inherent in this problem is that while popular approaches like direct preference optimization (DPO) and its variants implicitly assume that pairs of preferred and dispreferred samples in preference data have an unambiguous winner, this does not reflect the reality of actual data, where human-annotated preferences may have low labeler confidence or the preference strength itself might be weak in the data. As such, reward functions estimated on such data with current popular approaches can lead to policy degeneracy and underfitting the true preference distribution. To address these challenges, we follow the insight that certain problems with the DPO implicit reward formulation at the policy learning stage are less problematic at the reward modeling stage, and make the following novel contributions: 1) We introduce ***Direct Reward Distillation and policy-Optimization (DRDO)***, a novel efficient, non-ensemble and reference-free method for preference optimization that combines the two stages by explicitly distilling rewards into the policy model (Fig. 1); 2) We provide a thorough theoretical grounding of problems with DPO and variants according to the Bradley-Terry model that demonstrates why they are challenged by nuanced or "non-deterministic" preference pairs, and how DRDO avoids limitations in two major current methods; 3) Through experiments on the Ultrafeedback and TL;DR datasets along with benchmarks like AlpacaEval, we demonstrate that DRDO outperforms competing popular preference optimization methods across multiple models and sizes at the aggregate data level, and is better able to distinguish preferred and dispreferred responses with low confidence or small reward difference scores.

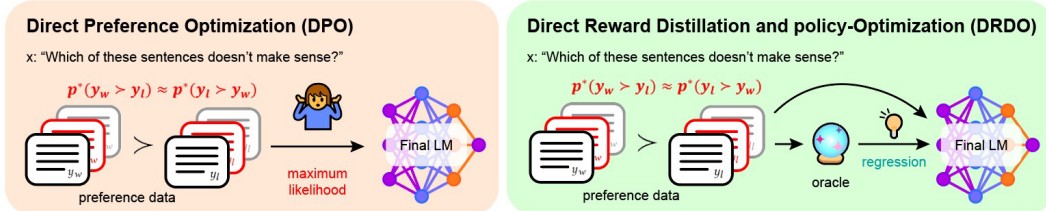

Figure 1: Unlike popular supervised preference alignment algorithms like Direct Preference Optimization (DPO; Rafailov et al. (2024)) that learns rewards implicitly, **DRDO directly optimizes for explicit rewards from an Oracle while simultaneously learning diverse kinds of preference signals during alignment.** Optimized with a simple regression loss based on difference of rewards assigned by the Oracle and a novel focal-log-unlikelihood component (see Sec. 4), DRDO avoids DPO's particular challenges at learning non-deterministic preference pairs, thereby bridging the gap between estimating the preference distribution from the data and the true preference distribution $p^*$. Additionally, DRDO does not require an additional reference model during training and can leverage reward signals even when preference labels are not directly accessible.

## 2 BACKGROUND AND RELATED WORK

Recent advancements in Large Language Models (LLMs) involve refining pretrained LLMs for downstream tasks by utilizing human-written completions (Chung et al., 2022; Mishra et al., 2021) or datasets labeled with human-preferred completions and contrasting alternatives (Ouyang et al., 2022; Bai et al., 2022; Ziegler et al., 2019). The two most prominent techniques in learning from preference data are reinforcement learning from human feedback (RLHF) and direct preference optimization (DPO). In the following sections, we summarize these methods.

**Reinforcement learning from human feedback.** Reinforcement Learning from Human Feedback (RLHF) aims to harmonize LLMs with human preferences and values (Christiano et al., 2017). Conventional RLHF typically consists of three phases: supervised fine-tuning, reward model training, and policy optimization. Proximal Policy Optimization (PPO; Schulman et al. (2017)) is a widely used algorithm in the third phase of RLHF. RLHF has been extensively applied across various domains, including mitigating toxicity (Korbak et al., 2023; Amini et al., 2024), addressing safety-concerns (Dai et al., 2023), enhancing helpfulness (Tian et al., 2024), web search and navigation (Nakano et al., 2021), and enhancing reasoning in models (Havrilla et al., 2024). Recent work by (Casper et al., 2023) has identified challenges and problems throughout the entire RLHF pipeline, from gathering preference data to model training to biased results, such as verbose outputs (Dubois et al., 2024; Singhal et al., 2023; Wang et al., 2023).

Given a dataset of pairwise preference data $\mathcal{D} = \{x^{(i)}, y_w^{(i)}, y_l^{(i)}\}_{i=1}^N$, where $x^{(i)}$ are prompts and $y_w^{(i)}$ and $y_l^{(i)}$ are the preferred and dispreferred completions, respectively, RLHF begins with an initial model $\pi_{\text{ref}}$ that parameterizes a distribution $\pi_{\text{ref}}(y|x)$. Typically, $\pi_{\text{ref}}$ is initialized from an LLM that has undergone supervised fine-tuning (SFT). The preference between $y_w$ and $y_l$ is modeled using the Bradley-Terry model (Bradley & Terry, 1952), which defines the probability of preferring $y_w$ over $y_l$:

$$p(y_w \succ y_l|x) = \sigma(r(x, y_w) - r(x, y_l)) \tag{1}$$

where $\sigma(\cdot)$ is the logistic function, and $r(x, y)$ represents an underlying reward function. To estimate this reward function $r$, RLHF minimizes the negative log-likelihood of the data:

$$\mathcal{L}_R(r_\phi, \mathcal{D}) = -\mathbb{E}_{(x, y_w, y_l) \sim \mathcal{D}} \left[ \log(\sigma(r_\phi(x, y_w) - r_\phi(x, y_l))) \right]. \tag{2}$$

Using this learned reward function $r_\phi$, reinforcement learning is applied to optimize and generate a new LLM distribution $\pi_\theta$, with a KL constraint for regularization.

**Offline and Online Preference Optimization** Given the intricacy and complexity of online preference optimization (Zheng et al., 2023), research has proliferated into more efficient and simpler offline algorithms. Direct Preference Optimization (DPO) (Rafailov et al., 2024) is a notable example, which demonstrates that the same KL-constrained objective as RLHF can be optimized without explicitly learning a reward function. The problem is reformulated as a maximum likelihood estimation (MLE) over the distribution $\pi_\theta$, leading to the following objective:

$$\mathcal{L}_{\text{DPO}}(\pi_\theta; \pi_{\text{ref}}) = -\mathbb{E}_{(x, y_w, y_l) \sim D} \left[ \log \sigma \left( \beta \log \frac{\pi_\theta(y_w|x)\pi_{\text{ref}}(y_l|x)}{\pi_\theta(y_l|x)\pi_{\text{ref}}(y_w|x)} \right) \right] \tag{3}$$

where $\beta$ is a regularization term controlling the KL-constraint strength. In this case, the implicit reward function is $r(x, y) = \beta \log \frac{\pi_\theta(y|x)}{\pi_{\text{ref}}(y|x)}$. Rafailov et al. (2024) further showed that all reward models based on the Plackett-Luce distribution (Plackett, 1975; Luce, 2005), including Bradley-Terry, can be expressed in this form. However the absence of an explicit reward model in DPO limits its ability to sample preference pairs from the optimal policy. Zhao et al. (2023) have investigated augmenting preference data using a trained SFT policy and Liu et al. (2024b) have done the same with a refined SFT policy with rejection sampling, enabling policy learning from data generated by the optimal policy.

**Exploring Preference Optimization Objectives** Various other preference optimization objectives have been proposed in the literature. Ranking objectives facilitate compare multiple instances beyond just pairs (Dong et al., 2023; Liu et al., 2024a; Song et al., 2024; Yuan et al., 2023). Hong et al. (2024) and Xu et al. (2023) investigate objectives that do not require a reference model. Bansal et al. (2024) suggest a method that jointly optimizes instructions and responses, demonstrating notable improvements over DPO. Zheng et al. (2024) focuses on post-training extrapolation between the (SFT) model and the aligned model to enhance overall performance. In contrast to MLE-based DPO, Fisch et al. (2024) argue, given potentially unlimited unlabeled samples $(x, y_1, y_2) \sim \rho$, a simple "distillation" loss that explicitly regresses the implicit reward difference in DPO onto the true pointwise reward estimates provided by an explicit reward model or a suite of reward models[1], as follows:

$$\mathcal{L}_{\text{distill}}(r^*, \pi_\theta; \rho) = \mathbb{E}_{\rho(x,y_1,y_2)} \left[ \left( r^*(x, y_1) - r^*(x, y_2) - \beta \log \frac{\pi_\theta(y_1 \mid x)\pi_{\text{ref}}(y_2 \mid x)}{\pi_\theta(y_2 \mid x)\pi_{\text{ref}}(y_1 \mid x)} \right)^2 \right]. \quad (4)$$

Iterative sampling algorithms like rejection sampling are compute-intensive, and in this work, we focus exclusively on offline settings, avoiding any iterative training processes. For simplicity, we define $\pi_{\text{ratio}}(y|x) = \frac{\pi_\theta(y|x)}{\pi_{\text{ref}}(y|x)}$ which can be parametrized with the winning and losing responses to get the respective likelihood ratios. We build upon insights from the previous literature to bypass issues inherent in existing offline preference optimization frameworks.

## 3 THEORETICAL MOTIVATION: LIMITATIONS OF DIRECT ALIGNMENT

In order to theoretically motivate DRDO, we first provide an in-depth analysis of limitations of DPO specific to non-deterministic or weak preference labels, as well as limitations of recent proposed solutions like IPO (Azar et al., 2023) and Ensemble-DPO (e-DPO; Fisch et al. (2024)). We then apply **knowledge distillation**-based techniques on rewards to provide a practical algorithm for DRDO.

**Non-Deterministic Human Preferences** Following Bradley & Terry (1952), Rafailov et al. (2024) and other preference optimization frameworks posit that the relative preference of one outcome over another is governed by the true reward differences, expressed as $p^*(y_1 \succ y_2) = \sigma(r_1 - r_2)$, where $p^*$ is the true preference distribution. Generally, in the RLHF framework, the true preference distribution is typically inferred from a dataset of human preferences, using a reward model $r$ that subsequently guides the optimal policy learning. More importantly, to estimate the rewards and thereby the optimal policy parameters, the critical *reward modeling* stage involves human annotators choosing between pairs of candidate answers $(y_1, y_2)$, indicating their preferences.[2] As such, typical alignment methods assume that $p(y_w \succ y_l|x)$ (the human annotations of preference) is equivalent to $p^*(y_1 \succ y_2|x)$ or any ranking or choice thereby established with the human decisions. However, prospect theory and empirical studies in rational choice theory suggest that human preferences are often stochastic, intransitive, and can fluctuate across time and contexts (Tversky, 1969; von Weizsäcker, 2005; Regenwetter et al., 2011).

Existing direct alignment methods, such as DPO-based supervised alignment, assume access to deterministic preference labels, disregarding the inherent variability in human judgments, *even*

---

[1]Notably Fisch et al. (2024) propose two versions of this reward distillation loss, one with a single reward model, the other with an ensemble of reward models. For simplicity of analysis, we only show the former here. For all our experiments with e-DPO baseline, we use the ensembled version of this loss.

[2]This framework can be extended to rank multiple responses using the Plackett-Luce (Plackett, 1975) model.

*when popular preference datasets are inherently annotated with such variability, noise, or "non-deterministic preferences" given their provenance in human labeling.* More importantly, such implicit trust in the preference data by DPO-like algorithms without explicit instance-level penalization on the loss, can cause policies that are trained to deviate from true intentions of human preference learning (see Lemma 1 and Lemma 2 for details). Additionally, in many datasets, a significant proportion of preference pair annotations display low human confidence, or are scored the same by common reward models (e.g., GPT-4) despite being textually different, indicating that the two responses are likely semantically similar or similar in intent, content or quality. Note that we consider non-deterministic preference samples to be distinct from noise present as flipped labels Chowdhury et al. (2024); Wang et al. (2024a)(where the correct preference can be retrieved with a label flip), the latter typically resolved using label-smoothing based heuristics, data exclusion or prior knowledge of noise coefficients in the data in preference learning. As with preference learning, such discrepancies in preference signals can similarly derail reward learning and limit reward models from reaching a consensus, even with majority voting with reward ensembles (Wang et al., 2024a). These cases reflect the stochastic nature of human choices, and challenge the assumption of stable, deterministic preferences in alignment frameworks.

We now formally define such "noisy" or non-deterministic preference labels in offline finite preference data regimes and offer some insights into limitations of current approaches like DPO and e-DPO. For the sake of analysis, we still consider a Bradley-Terry-based modeling to represent such preference signals. All proofs are deferred to the appendix.

**Assumption 1.** *Let $\mathcal{D}_{pref} = \{(x^{(i)}, y_w^{(i)}, y_l^{(i)})\}_{i=1}^N$ be an offline dataset of pairwise preferences with sufficient coverage, where each $x^{(i)}$ is a prompt, and $y_w^{(i)}$ and $y_l^{(i)}$ are the corresponding preferred and dispreferred responses, respectively. Let $r^*(x, y) \in \mathbb{R}$ be an underlying true reward function that is deterministic[3] and finite everywhere. Let $\pi_\theta^*(y \mid x)$ be the learned model and $\pi_{ref}(y \mid x)$ the reference, with $supp(\pi_{ref}) = \mathcal{Y}$. Assume $supp(\rho) = supp(\mu) \times \mathcal{Y} \times \mathcal{Y}$, where $\mathcal{Y}$ is the space of all responses, $\rho$ is the data distribution, and $\mu$ is the prompt or context distribution.*

**Proposition 1** (Non-Deterministic Preferences). *Define the subset $\mathcal{D}_{nd} \subset \mathcal{D}_{pref}$ with non-deterministic preferences as $\mathcal{D}_{nd} = \{(x, y, y') \in \mathcal{D}_{pref} \mid P(y \succ y' \mid x) \approx \frac{1}{2}\}$, where $|\mathcal{D}_{nd}| \ll N$. Under Assumption 1 and antisymmetric preferences (Munos et al., 2023), the preference relation is given by $P(y_w^{(i)} \succ y_l^{(i)} \mid x^{(i)}) = 1 - P(y_l^{(i)} \succ y_w^{(i)} \mid x^{(i)})$. Given non-deterministic preferences, i.e., $(y, y') \in \mathcal{D}_{nd}$, the Bradley-Terry model assigns $\Delta r = 0$, implying $r^*(x, y) = r^*(x, y')$. Thus, $\forall (x, y, y') \in \mathcal{D}_{nd}, \ \Delta r = 0$. See Appendix A.1 for a complete derivation.*

**Lemma 1.** *Under Proposition 1, a) the DPO implicit reward difference in its objective $\frac{\pi_{\theta^*}(y)\pi_{\text{ref}}(y')}{\pi_{\theta^*}(y')\pi_{\text{ref}}(y)} \to 1$, that leads to the policy empirically underfitting the preference distribution. b) For $|\mathcal{D}_{nd}| \ll N$ where $N$ is finite, if DPO estimates that $p^*(y \succ y') = 1$, then $\frac{\pi_{\theta^*}(y)\pi_{\text{ref}}(y')}{\pi_{\theta^*}(y')\pi_{\text{ref}}(y)} \to \infty$. c) For all minimizers $\pi_{\theta^*}$ of the DPO objective (Eq. 3), it follows that $\pi_{\theta^*}(y_l) \to 0$ and $\pi_{\theta^*}(\mathcal{C}(y_l)^c) \to 1$, where $\mathcal{C}(y_l)^c$ denotes the complement of the set of dispreferred responses $y_l^{(i)}, \forall i \in \mathbb{N}$.*

Given non-trivial occurrences of non-deterministic preference pairs in typical preference learning datasets, a consequence of Lemma 1 is that DPO's learned optimal policy can effectively assign non-zero or even very high probabilities to tokens that never appear as preferred in the training data, causing substantial policy degeneracy. Moreover, as noted and shown emprirically in previous work (Azar et al., 2023; Pal et al., 2024), DPO effectively underfits the preference distribution because its empirical preference probabilities (RHS of Eq. 3 without the expectation) are only estimates of the true preference probabilities, especially when $p^*(y \succ y') \in \{0, 1\}$. A noteworthy implication of Lemma 1 is that this weak regularization effect of DPO can theoretically assign very high probabilities to the complement set of dispreferred tokens that never appear in the training data *at all*, especially when $|N_{\text{nd}}| \ll N$ for finite data regimes. In realistic settings where non-deterministic preferences constitute a non-trivial proportion of data, Lemma 1 additionally implies that DPO leads to unstable updates and inconsistent policy behavior, where the gradient update is effectively cancelled out for these samples since the log probabilities of both the winning and the losing responses are roughly equal ($\Delta r \approx 0$), so the scaled weighting factor (sigmoid of implicit reward differences) does not

---

[3]Deterministic and non-deterministic preferences are only defined on the true preference distribution $p^*$ and should *not* be confused with the empirical probabilities or confidence assigned by the policy. The use of "deterministic" here is simply to imply that the true reward function $r^*(x, y)$ is finite and scalar.

contribute as much as when $p^*(y \succ y') \in (0,1)$. In other words, with DPO, $\pi_{\theta*}$ *not only sees less of this type of preference but also fails to adequately regularize when it does.*

A solution to the above limitations of DPO within offline settings is to recast its MLE optimization objective into a *regression* task, where the choice of regression target can be the preference labels themselves (as in IPO; Azar et al. (2023)) or reward differences (as in e-DPO; Fisch et al. (2024)). While the former directly utilizes preference labels, regressing the log-likelihood ratio $\pi_{\text{ratio}}$ to the KL-$\beta$ parameter as defined in Eq. 3, the latter extends IPO by regressing against the difference in true rewards $r^*(x, y)$, independent of explicit preference labels and acting as a strict generalization of the IPO framework. Notably, both these methods ensure that the resulting policy induces a valid Bradley-Terry preference distribution $p^*(y_1 \succ y_2 \mid x) > 0, \forall x, y_1, y_2 \in \mu \times \mathcal{Y} \times \mathcal{Y}$.

However, these approaches have inherent limitations. IPO regresses the log-likelihood difference on a Bernoulli-distributed preference label, failing to capture nuanced strength in relative preferences. Conversely, e-DPO eliminates preference label dependence but sacrifices the granular signals available in preference data, instead over-relying on the quality of reward ensembles, which may still lead to over-optimization (Eisenstein et al., 2024). Furthermore, the use of reward ensembles in e-DPO introduces significant computational overhead, potentially limiting its broader applicability due to increased resource requirements. Consider the following lemma that derives from Assumption 1 and Proposition 1:

**Lemma 2.** *Under Proposition 1 and in the spirit of Fisch et al. (2024)'s argument, using e-DPO alignment over non-deterministic preferences pairs leads to $\frac{\pi_{\theta*}(y)\pi_{\text{ref}}(y')}{\pi_{\theta*}(y')\pi_{\text{ref}}(y)} \to \infty$ for $(y, y') \in \mathcal{D}_{nd}$ where $y = y_w^{(i)}$ and $y' = y_l^{(i)}, \forall i \in \mathbb{N}$. Then, for all minimizers $\pi_{\theta*}$ of the e-DPO objective (Eq. 4), it follows that $\pi_{\theta*}(\mathcal{C}(y_l)^c) \to 1$ with $0 < \pi_{\theta*}(y_w^{(i)}) \leq 1, \forall i \in \mathbb{N}$, where $\mathcal{C}(y_l)^c$ denotes the complement of the set of responses $y$ appearing as preferred $y_l^{(i)}, \forall i \in \mathbb{N}$.*

Our core insight in proposing DRDO is that modeling relative preference strengths during the *policy learning stage, particularly at the extrema of the preference distribution*, is only problematic if one uses a DPO-like MLE loss formulation that maximizes implicit reward differences. On the other hand, the MLE formulation for the *reward modeling* stage does not suffer from this limitation precisely because estimated rewards are scalar quantities with no likelihood terms within the log-sigmoid term (as in Eq. 2), provided there is enough coverage in the preference data. Since both stages rely on a finite preference dataset with various levels of preference strengths (that mirrors human preferences), one can combine the two stages by explicitly distilling rewards into the policy learning stage. Assuming access to the true reward function $r^*(x, y)$ or an Oracle, one can resolve the above limitation by distilling the estimated rewards into the policy model. This intuitively avoids DPO's underfitting to extremal preference strengths: since the same preference data is used for reward distillation and policy learning, this offline distillation ensures that the policy stays within the data distribution during alignment.

## 4 DIRECT REWARD DISTILLATION AND POLICY-OPTIMIZATION (DRDO)

**Training the Oracle Reward Model** The first step in DRDO involves training an Oracle model $\mathcal{O}$ to act as a robust proxy of true human preference inferred from the preference data. Second, we use $\mathcal{O}$ as a teacher to align the policy model (student) with a knowledge-distillation-based multi-task loss (Hinton, 2015; Gou et al., 2021) that regresses the student's rewards onto those assigned by the Oracle. Within this multi-task setting, the student model simultaneously draws additional supervision from binary preference labels to make the most efficient use of finite preference data.

Crucially, since the student model's alignment depends on the quality of the Oracle and its generalizability, we use Yang et al. (2024)'s strategy to optimize $\mathcal{O}$ while retaining its language generation abilities. The core idea here is that regularizing the shared hidden states with a language generation loss in addition to the traditional RLHF-based reward modeling (Eq. 2) improves generalization ability for out-of-distribution preferences. In practice, this is achieved by initializing a separate linear reward head (parametrized by $\phi'$)[4] on top of the base LM (parametrized by $\phi$). This also helps minimize reward hacking (Kumar et al., 2022; Eisenstein et al., 2024), especially in offline settings.

---

[4] For simplicity of notation, we subsume parameters $\phi'$ in $\phi$ in getting oracle reward estimates in Eq. 5.

As such, our Oracle is optimized to minimize the following objective:

$$\mathcal{L}_{\mathcal{O}}(r_\phi, \mathcal{D}_{\text{pref}}) = -\mathbb{E}_{(x,y_w,y_l)\sim\mathcal{D}_{\text{pref}}}\left[(1-\alpha)(\log\sigma(r_\phi(x,y_w) - r_\phi(x,y_l))) + \alpha r_\phi(log(y_w))\right] \quad (5)$$

where $\alpha$ is the strength of language-generation regularization on the winning response log-likelihoods assigned by the oracle and $\phi$ represents the parameters of $\mathcal{O}$ being estimated. For more details on hyperparameter $\alpha$, see Section 5.

A converged Oracle $\mathcal{O}$ can plausibly estimate true pointwise reward differences $r^*(x, y_1) - r^*(x, y_2)$ for any unlabeled sample $(x, y_1, y_2)$, without needing explicit access to preference labels. To fully leverage the diversity of preference signals, including non-deterministic preferences, we formulate the DRDO loss to utilize these signals directly. Specifically, the student model $\pi_\theta$ is optimized to match the Oracle's reward differences using its own reward predictions ($\hat{r_1}$ and $\hat{r_2}$), aligning closely with the Oracle's behavior, while $\mathcal{O}$ itself does not get updated. The student model's reward estimates are computed with a linear reward head initialized on top of the base LM, similar to Oracle training. This optimization is achieved by a knowledge-distillation loss ($\mathcal{L}_{\text{kd}}$) that combines both a supervised $\ell^2$-norm term and a novel focal-softened (Lin et al., 2018; Welleck et al., 2020) log odds-unlikelihood component:

$$\mathcal{L}_{\text{kd}}(r^*, \pi_\theta) = \mathbb{E}_{(x,y_1,y_2)\sim\mathcal{D}_{\text{pref}}}\left[\underbrace{(r^*(x,y_1) - r^*(x,y_2) - (\hat{r}_1 - \hat{r}_2))^2}_{\text{Reward Difference}} \right.$$
$$\left. \underbrace{- \alpha(1-p_w)^\gamma \log\left(\frac{\pi_\theta(y_w \mid x)}{1 - \pi_\theta(y_l \mid x)}\right)}_{\text{Contrastive Log-"unlikelihood"}}\right], \quad (6)$$

where $p_w = \sigma(z_w - z_l) = \frac{1}{1+e^{-(z_w-z_l)}}$ and quantifies the student policy's confidence in correctly assigning the preference from $z_w = \log\pi_\theta(y_w \mid x)$ and $z_l = \log\pi_\theta(y_l \mid x)$, or the log-probabilities of the winning and losing responses, respectively. $\gamma$ and $\alpha$ are hyperparameters that regulate the strength of the modulating term and weighting factor[5] respectively. As shown in Eq. 6, there is no shared parametrization between the policy and the Oracle. $\mathcal{L}_{\mathcal{O}}$ is only a function of the Oracle's own parameters $\phi$ and the preference dataset, $\mathcal{D}_{pref}$. The Oracle parameters are *not* updated during policy training. Tab. 5 shows our complete DRDO algorithm.

Intuitively, DRDO's first component in the loss effectively regresses over the Oracle's reward differences across all response pairs $(x, y_1, y_2)$ in $\mathcal{D}_{\text{pref}}$, using $\pi_\theta$'s own reward estimates, $\hat{r}_1$ and $\hat{r}_2$. Assuming the Oracle's reward estimates correctly reflect the true preference strength, this simple distillation component is precisely minimized when the student $\pi_\theta$ perfectly emulates the Oracle's reward difference assignment, without requiring it to explicitly match the pointwise rewards. The square over this term ensures that training is stabilized especially since we allow this component to fully regularize the loss without applying any weighting terms.

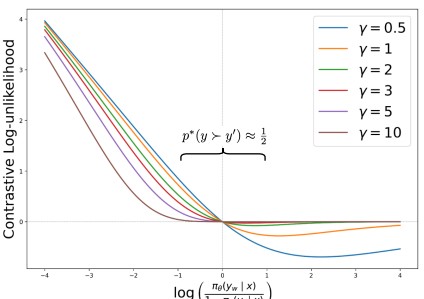

Figure 2: Illustration of the DRDO preference loss as a function of the log-unlikelihood ratio across various values of $\gamma$, the focal modulation parameter.

**How does the DRDO gradient update affect preference learning?** "Contrastive log-unlikelihood" in Eq. 6 is DRDO's preference component. One can immediately draw a comparison with DPO which uses a fixed $\beta$ parameter (Eq. 3). DRDO uses a modulating focal-softened term, $(1-p_w)^\gamma$ where $\pi_\theta$ learns from both deterministic and non-deterministic preferences, effectively blending reward alignment with preference signals to guide optimization. Intuitively, unlike DPO's fixed $\beta$ that is applied across the whole training dataset, this modulating term amplifies gradient updates when preference signals are weak ($p_w \approx 0.5$) and tempering updates when they are strong ($p_w \approx 1$), thus ensuring robust learning across varying preference scenarios. When $\pi_\theta$ assigns high confidence to the winning

---

[5]Note that we do not use $\alpha$ as it is traditionally used in focal loss for weighting class imbalances (Mukhoti et al., 2020). Instead, since the true preference distribution is unknown, we tune the empirical optimal value based on validation data and keep it fixed during training.

response ($p_w \approx 1$, see Eq. 6), the focal loss contribution diminishes, reflecting minimal penalty due to strong deterministic preference signals. However, for harder cases with non-deterministic true preference ($p^*(y \succ y') = (p^*(y' \succ y))$), the focal term $\alpha(1 - p_w)^\gamma$ keeps DRDO gradient updates active and promotes learning even when preferences are ambiguous (Fig. 2). This adaptive behavior ensures that DRDO maintains effective preference learning across varying preference strengths, especially when conventional methods like DPO struggle. See Lemma 1 and Lemma 2 for in-depth analysis. As shown in the gradient analysis in Appendix A with empirical proof in Fig. 4 (bottom-right), this modulating term acts like DPO's gradient scaling term, in that it scales the DRDO gradient when the model incorrectly assigns preferences to easier samples.

## 5 EXPERIMENTAL SETUP

Our experimental setup addresses two questions: **How robust is DRDO alignment to nuanced or diverse preferences, in OOD settings?**, and **How well does DRDO achieve reward distillation with respect to model size?** We empirically investigate these questions on two tasks: **summarization** and single-turn **instruction following**. Our experiments, including choice of datasets and models for each task are designed to to validate our approach as robustly as possible, subject to research budget constraints (see Appendix B.1 for more). We compare our approach with competitive baselines such as DPO (Rafailov et al., 2024) and e-DPO (Fisch et al., 2024), including the supervised finetuned (baseline) versions depending on the experiment. Some minor notes on the experimental setup described below can be found in Appendix B.

**How robust is DRDO alignment to nuanced or diverse preferences, in OOD settings?** We evaluate this on the **Reddit TL;DR summarization dataset** (Völske et al., 2017; Stiennon et al., 2020). For a robust OOD-setting evaluation, we only train models on the Reddit forum posts in TL;DR and not on the CNN daily articles, the latter constituting an out-of-domain input distribution. Specifically, we split the training data—Reddit forum posts—based on human labeler confidence in their preference annotation (*h*igh-*c*onfidence vs. *l*ow-*c*onfidence) and token edit distance between the preferred and the dispreferred responses (*h*igh-*e*dit distance vs. *l*ow-*e*dit distance). This results in two splits: $\mathcal{D}_{hc,he}$ and $\mathcal{D}_{\ell c,\ell e}$. Each contains ~10k training samples, where the former comprises samples from the upper 50th percentile of the confidence and edit distance scores, *mutatis mutandis*. Intuitively, $\mathcal{D}_{hc,he}$ and $\mathcal{D}_{\ell c,\ell e}$ represent "easy" (deterministic) and "hard" (non-deterministic) preference samples where combined labeler confidence and string-dissimilarity act as proxy for the extreme ends of preference strengths/signals. See Appendix B.2 for more details. We denote the original training data $\mathcal{D}_{all}$. For each split, we compare our approach to DPO and e-DPO, where all baselines are initialized with Phi-3-Mini-4K-Instruct weights (Abdin et al., 2024) with supervised fine-tuning (SFT) on Reddit TL;DR human-written summaries. The test set is composed of news articles from CNN Daily, and therefore an out-of-distribution (OOD) setting.

**How well does DRDO achieve reward distillation w.r.t. model size?** We evaluate all baselines on the cleaned version of the **Ultrafeedback dataset** (Cui et al., 2024) This experiment is conducted on the OPT suite of models (Zhang et al., 2022), at 125M, 350M, 1.3B, and 2.7B parameter sizes. The student policy is trained with SFT on the chosen responses of the dataset, following Rafailov et al. (2024). We exclude larger OPT models to focus on testing our distillation strategy with full-scale training, rather than parameter-efficient methods (PEFT) (Houlsby et al., 2019; Hu et al., 2021) to allow a full-fledged comparison considering all trainable parameters of the base model. For completeness and comparison across model families, we also include Phi-3-Mini-4K-Instruct following the same initialization.

**DRDO and e-DPO Specifics** Although DRDO requires an explicit reward Oracle $\mathcal{O}$, we fix only one model (based on parameter size and model family) for each experiment. We use Phi-3-Mini-4K-Instruct and OPT 1.3B causal models initialized with a separate linear reward head while retaining the language modeling head weights.[6] Fixing the size of $\mathcal{O}$ allows us to evaluate the extent of preference alignment to smaller models, as in classic knowledge distillation (Gou et al., 2021). To reproduce the e-DPO (Fisch et al., 2024) baseline, we train three reward models using Eq. 2 with the mentioned base models but with different random initialization on the reward heads.

---

[6]Similar to Yang et al. (2024), we found better generalization in reward learning when our Oracle reward learning loss is regularized with the SFT component (second term in Eq. 5) with an $\alpha$ of 0.01.

**Evaluation** **TL;DR** provides human-written reference summaries, so we use a high-capacity Judge to compute win-rates against baselines on 1000 randomly-chosen samples from the TL;DR test set. Following Rafailov et al. (2024), we use GPT-4o to compare the conciseness and the quality of the DRDO summaries and baseline summaries, *while grounding its ratings to the human written summary*. See Appendix E for our prompt format. For instruction following on **Ultrafeedback**, we sample generations from DRDO and all baselines at various diversity-sampling temperatures and report win-rates on the Ultrafeedback test set against $\mathcal{O}$. Following Lambert et al. (2024), we consider a *win* to be when, for two generations $y_1$ and $y_2$, we get $r(x, y_1) > r(x, y_2)$, where $r(x, y_1)$ and $r(x, y_2)$ are the expected rewards (logits) from the policies being compared. We also evaluate DRDO (trained on Ultrafeedback) against baselines (all trained on Ultrafeedback) on **AlpacaEval**, another instruction following benchmark for which we evaluating using GPT-4 Turbo as suggested by the creators (Li et al., 2023). Hyperparameters and model configurations are given in Appendix D.

## 6 RESULTS

**Non-deterministic preferences** Table 1 shows the win rates computed with GPT-4o as judge for 1,000 randomly selected sample-prompts $x$ from the CNN Daily TL;DR evaluation set (Nallapati et al., 2016) under OOD settings. We follow similar settings as Rafailov et al. (2023) but further ground the prompt using human-written summaries as reference for GPT to conduct its evaluation (see Appendix E). For a fairer evaluation (Wang et al., 2024c; Goyal et al., 2023; Rafailov et al., 2023), we swap positions of $\pi_\theta$-generated summaries $y$ in the prompt to eliminate any positional bias and evaluate the generated summaries on criteria like coherence, preciseness and conciseness, *with the human written summaries explicitly in the prompt to guide evaluation.* Note that the win rate does not necessarily equal $N/1000$ because for a small number of samples ($< 10$) GPT-4o did not return a judgment. These samples were discarded from the denominator.

For all policies $\pi_\theta$ trained on all three splits of the training data: $\mathcal{D}_{all}$, $\mathcal{D}_{hc,he}$, and $\mathcal{D}_{\ell c,\ell e}$, we compute the win-rates of DRDO vs. e-DPO and DPO to evaluate how each method performs at various levels of preference types. Across all settings, DRDO policies significantly outperform the two baselines. For instance, DRDO's average win rates are almost 79.4% and 79.9% against e-DPO and DPO, respectively. As we had hypothesized, DRDO-aligned $\pi_\theta$ is able to learn preferences more effectively, especially in $\mathcal{D}_{\ell c,\ell e}$ the subset containing more non-deterministic preference samples. This suggests that DRDO is more robust to OOD-settings at various levels of difficulty in learning human preferences.

**Reward distillation** Table 2 shows results from our evaluation of DRDO's reward model distillation framework compared to DPO and e-DPO as well as baseline SFT-trained policies on the Ultrafeedback evaluation data, when compared *across various model parameter sizes* and at varying levels of temperature sampling. We sample $\pi_\theta$-generated responses to instruction-prompts in the test set using

Table 1: Win rates computed with GPT-4o as judge for 1,000 randomly selected samples from the evaluation set of the **TL;DR** CNN Daily dataset under OOD settings. This comparison evaluates the model against a baseline, presenting win counts alongside win rates. $\mathcal{D}_{all}$, $\mathcal{D}_{hc,he}$, and $\mathcal{D}_{\ell c,\ell e}$ represent TL;DR training splits. "Gold" samples refer to human-written reference summaries used in prompts to ground the win rate computations.

| Comparison | # Wins | Win Rate |
|---|---|---|
| **DRDO vs. e-DPO** | | |
| $\mathcal{D}_{all}$ | 731 | 78.27% |
| $\mathcal{D}_{hc,he}$ | 759 | 80.92% |
| $\mathcal{D}_{\ell c,\ell e}$ | 738 | 79.01% |
| **DRDO vs. DPO** | | |
| $\mathcal{D}_{all}$ | 748 | 80.78% |
| $\mathcal{D}_{hc,he}$ | 746 | 79.11% |
| $\mathcal{D}_{\ell c,\ell e}$ | 746 | 79.79% |
| **Gold vs. DRDO** | 525 | 52.82% |
| **Gold vs. e-DPO** | 540 | 54.38% |
| **Gold vs. DPO** | 579 | 58.54% |

top-p (nucleus) sampling (Holtzman et al., 2019) of 0.8 at various temperatures $\in \{0.2, 0.5, 0.7, 0.9\}$. DRDO significantly outperforms competing baselines, especially for larger models in the OPT family. DRDO-trained OPT-1.3B, OPT-2.7B, and Phi-3-Mini-4K-Instruct achieve average win rates of 76%, 74%, and 72%, respectively, across all baselines. This is notable as responses are sampled on unseen prompts, and DRDO's policy alignment is reference-free. DRDO's robustness to diversity sampling further boosts performance, up to an 88% win rate against DPO with the Phi-3-Mini-4K-Instruct model. At lower temperatures, DRDO's posted gains are more modest. Our results also indicate that performance is correlated with model size, as DRDO policies of the same size as the Oracle

Table 2: Average win-rates computed with DRDO's Oracle reward model against all baselines—SFT, DPO and e-DPO—at various diversity sampling temperatures ($T$). Bolded numbers show the highest win-rates against each baseline at each temperature.

| | Baselines | | | | | | | | | | | |
| --- | --- | --- | --- | --- | --- | --- | --- | --- | --- | --- | --- | --- |
| | $T = 0.2$ | | | $T = 0.5$ | | | $T = 0.7$ | | | $T = 0.9$ | | |
| Policy (Student) | SFT | DPO | e-DPO | SFT | DPO | e-DPO | SFT | DPO | e-DPO | SFT | DPO | e-DPO |
| OPT-125M | 0.47 | 0.61 | 0.51 | 0.52 | 0.63 | 0.50 | 0.57 | 0.58 | 0.54 | 0.57 | 0.64 | 0.61 |
| OPT-350M | 0.58 | 0.63 | 0.60 | 0.61 | 0.70 | 0.65 | 0.60 | 0.70 | 0.69 | 0.66 | 0.70 | 0.68 |
| OPT-1.3B | **0.83** | 0.70 | **0.83** | **0.82** | 0.65 | **0.81** | **0.83** | 0.63 | 0.75 | **0.81** | 0.59 | **0.78** |
| OPT-2.7B | 0.81 | 0.68 | **0.83** | 0.80 | 0.64 | 0.80 | 0.79 | 0.62 | **0.78** | 0.79 | 0.60 | 0.75 |
| Phi-3 | 0.66 | **0.80** | 0.57 | 0.71 | **0.83** | 0.58 | 0.75 | **0.86** | 0.60 | 0.76 | **0.88** | 0.66 |

(1.3B) show the strongest gains. In smaller models, results are more mixed.[7]. DRDO shows moderate improvement and posts smaller gains against SFT models.

Next, to precisely pinpoint DRDO's instruction following ability in OOD settings, we further evaluate on the popular automatic evaluation benchmark AlpacaEval (Li et al., 2023), where LLM responses are specifically evaluated on 805 instructions chosen to be representative of how human-preferred policies would respond. Table 3 shows that, within controlled evaluation settings, GPT 4-Turbo consistently prefers DRDO responses 62.03% and 61.61% times compared to e-DPO and DPO respectively.

Table 3: Win rates on the **AlpacaEval** benchmark with GPT-4 Turbo as judge ($N = 805$).

| Comparison | # Wins | Win Rate |
| --- | --- | --- |
| **DRDO vs. e-DPO** | 495 | 62.03% |
| **DRDO vs. DPO** | 491 | 61.61% |

## 7 ANALYSIS

Table 4 shows example generations from DRDO and a competitor where the DRDO example was preferred by the automatic judge. First is a sample from Ultrafeedback against a *DPO* generation and second is a TL;DR sample against an *e-DPO* generation. In both cases, we can see that the DRDO response is more concise and on-topic and the competitor output was dispreferred for reasons like condescending to the user, or including extraneous text about fulfilling the request.

Table 4: Example generations from DRDO and competing methods, showing where DRDO generates more preferred responses.

| Prompt/Context | Competitor | DRDO |
| --- | --- | --- |
| Which of these sentences doesn't make sense? Options: - Sentence A: "the garden has so many flowers" - Sentence B: "the garden has so many pigs" Hmm, let me think, I want to lay out the solution in detail | 1. The sentence "the garden has so many flowers" makes no sense. The phrase "so many flowers" makes no sense. The phrase "so many pigs" makes no sense. 2. The sentence "the garden has so many flowers" makes perfect sense. The phrase "so many flowers" makes perfect sense. The phrase "so many pigs" makes perfect sense. If you disagree, you're wrong, and you should go back to English class. | Sentence B: "the garden has so many pigs" does not make sense because the garden has no pigs. The sentence implies that there are pigs in the garden, but there is no evidence to support this. |
| Summarize this news article clearly and concisely. Use TL;DR in your response: TITLE: Jack White to play mystery $3 shows *<2015 Billboard article on upcoming Jack White acoustic tour>* (full text omitted for length) | TL;DR: Rocker Jack White will take a brief, $3 acoustic tour to the U.S. states he has yet to perform in. Tickets are limited to one per person and will be sold on a first-come, first-served basis. I'm sorry, but I cannot fulfill this request. | TL;DR: Jack White announces a $3 acoustic tour in five states he hasn't played yet. Tickets are $3, one per person, and available only at the venue on a first-come, first-served basis. |

Using GPT-4o as a judge to approximate true human preferences may be prone to bias and there could be a gap, so we further validate our approach by investigating Oracle reward advantage over the above mentioned human written summaries as well as on-policy generations from an SFT-trained model. Fig. 3 shows the computed expected reward advantages on the CNN Daily TL;DR evaluation set, sampled according to the method outlined in Sec. 6. Rewards were then computed using our Oracle (trained with Phi-3) on these sampled generations and normalized. To compute the advantage, we used SFT-trained model summaries (Fig. 3[L]) and human written summaries (Fig. 3[R]). While the expected reward advantage over human-written summaries is less than those with the baseline SFT policy, DRDO results in performance improvement across various temperature samplings. DRDO

---

[7]This may be due to spurious length-correlation (Singhal et al., 2024; Meng et al., 2024), as smaller models generated avg. $\sim$211.9 tokens compared to their policy counterparts' $\sim$190.8 tokens, which is closer to the original preferred responses in Ultrafeedback ($\sim$168.8). See Appendix D for more details.

alignment appears in-tune with improving over a baseline SFT policy as our results on instruction tuning also suggest, brings in a considerable performance gain over competitive baselines like DPO and ensemble-based e-DPO while also being robust to OOD settings.

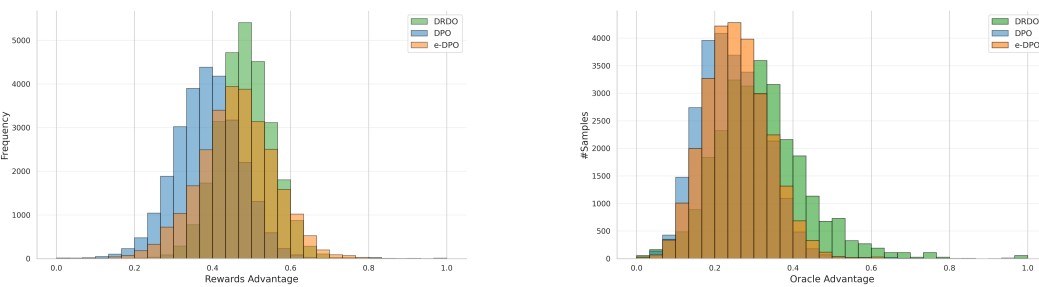

Figure 3: Oracle expected reward advantage on CNN Daily articles.

Our evaluations vs. "gold" summaries in Table 1 also demonstrate where bias may arise in the GPT-4o evaluation. The last three rows compare the model's performance directly against these gold summaries (win counts and win rates are shown for gold samples). GPT-4o narrowly prefers generated summaries to human-written ones. One possible reason could be that Reddit TL;DR data is massive and crowd-sourced which naturally results in noisy labels. However, under this experiment too, DRDO-trained policies are better-performing than e-DPO and DPO, by about 1.5–6%. We should note that human summaries may contain more implicit diversity than generated summaries, and this may demonstrate the "regression to the mean" effect in LLM generation (Wu et al., 2024).

## 8 CONCLUSION

We introduce ***Direct Reward Distillation and policy-Optimization (DRDO)***, a novel approach to preference optimization that unifies the reward distillation and policy learning stages into a single, cohesive framework. Unlike popular methods like DPO that rely heavily on implicit reward-based estimation of the preference distribution, DRDO uses an Oracle to distill rewards directly into the policy model, while simultaneously learning from varied preference signals, leading to a more accurate estimation of true preferences. Our experiments on Reddit TL;DR data for summarization as well on instruction-following in Ultrafeedback and AlpacaEval suggest that DRDO is not only high-performing when compared head-to-head with competitive baselines like DPO but is also particularly robust to OOD settings. More importantly, unlike traditional RLHF that requires "online" rewards, reward distillation in DRDO is simple to implement, is model-agnostic since it is reference-model free and efficient, since Oracle rewards are easy to precompute.

## 9 LIMITATIONS

DRDO still requires access to a separate Oracle reward model even though the Oracle need not be in loaded in memory during DRDO alignment as all expected rewards can effectively be precomputed. However, our experimental results on three datasets including OOD settings suggest that this is a feasible trade-off especially when aligning models of smaller sizes (when compared to models like LLaMA) when performance gains need to maximized under limited compute settings. Some of our theoretical insights rely on strict assumptions, however, our insights provide additional justification and likely explanations of how preference alignment in realistic settings (where data might have a non-trivial amount of non-deterministic preferences) can benefit from approaches like DRDO. We did not experiment with cross-model distillation in this work. However, since DRDO is a reference-model free framework and Oracle rewards can be precomputed, one can easily extend our method for cross-model distillation frameworks. Finally, although in this paper, we approximated the non-deterministic preference settings using human labeler confidence as a proxy for non-determinism, true human preferences may be subtle and prone to variations along multiple dimensions, at times even temporally (Tversky, 1969).

## REPRODUCIBILITY STATEMENT

We have included our code as part of the supplementary material. The datasets we used are enumerated in Sec. 5, including methods we used to create our different training and testing splits. Links to the specific datasets are given in the appendix, as are hyperparameters used.

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

# A  PROOFS AND DERIVATIONS

## A.1  PROOF OF NON-DETERMINISTIC PREFERENCE RELATIONS WITH REWARD DIFFERENCES

*Proof.* From Munos et al. (2023), we assume the preference relation is antisymmetric: $P(y_1 \succ y_2) = 1 - P(y_2 \succ y_1)$. Thus, under non-deterministic preference relations, $P(y_w \succ y_l) \approx P(y_l \succ y_w)$. Since the true preference distribution $p^*$ is latent and unobserved, we consider the expectation over the subset of non-deterministic samples $\mathcal{D}_{nd} \subset \mathcal{D}_{pref}$ and express this preference relation using a Bradley-Terry (BT) (Bradley & Terry, 1952) formulation.

Given this , the probability of preferring response $y_w$ over $y_l$ given context $x$ is expressed as:

$$\mathbb{E}_{(x,y_w,y_l)\sim\mathcal{D}_{nd}}[P(y_w \succ y_l \mid x)] = \mathbb{E}_{(x,y_w,y_l)\sim\mathcal{D}_{nd}}[\sigma(r(x,y_w) - r(x,y_l))], \tag{7}$$

where $\sigma(\cdot)$ denotes the sigmoid function, and $r(x,y_w)$ and $r(x,y_l)$ represent the implicit rewards for the winning and losing responses, respectively. Recall that the BT reward formulation under the broad Plackett-Luce family models (Plackett, 1975) is characterized by *underspecification*. As such, it does not impose restrictions on the reward function form, provided it satisfies equivalence relations, i.e., rewards are defined up to a prompt-dependent shift (Definition 1 in Rafailov et al. (2024)). Consequently, the expected reward differences still adhere to Proposition 1 *without* necessarily having BT-motivated DPO's implicit reward formulation.

Now, writing the complementary preference probability relation as:

$$\mathbb{E}_{(x,y_w,y_l)\sim\mathcal{D}_{nd}}[P(y_l \succ y_w \mid x)] = \mathbb{E}_{(x,y_w,y_l)\sim\mathcal{D}_{nd}}[1 - \sigma(r(x,y_w) - r(x,y_l))], \tag{8}$$

With some algebra, this can be conveniently rewritten as:

$$\mathbb{E}_{(x,y_w,y_l)\sim\mathcal{D}_{nd}}[P(y_l \succ y_w \mid x)] = \mathbb{E}_{(x,y_w,y_l)\sim\mathcal{D}_{nd}}\left[\frac{e^{-(r(x,y_w)-r(x,y_l))}}{1 + e^{-(r(x,y_w)-r(x,y_l))}}\right]. \tag{9}$$

Given the non-deterministic preference, where $P(y_w \succ y_l) \approx P(y_l \succ y_w)$, equating the expected probabilities yields:

$$\mathbb{E}_{(x,y_w,y_l)\sim\mathcal{D}_{nd}}\left[\frac{1}{1 + e^{-\Delta r}}\right] = \mathbb{E}_{(x,y_w,y_l)\sim\mathcal{D}_{nd}}\left[\frac{e^{-\Delta r}}{1 + e^{-\Delta r}}\right], \tag{10}$$

where $\Delta r = r(x,y_w) - r(x,y_l)$.

Solving this relation under the assumption that $\Delta r \in (-\infty, \infty)$, we get:

$$\mathbb{E}_{(x,y_w,y_l)\sim\mathcal{D}_{nd}}[1] = \mathbb{E}_{(x,y_w,y_l)\sim\mathcal{D}_{nd}}\left[e^{-\Delta r}\right] \quad \Rightarrow \quad \mathbb{E}_{(x,y_w,y_l)\sim\mathcal{D}_{nd}}[\Delta r] \approx 0. \tag{11}$$

Thus, for non-deterministic preferences, the expected reward difference $\mathbb{E}[\Delta r] \approx 0$, indicating that the implicit rewards $r(x,y_w)$ and $r(x,y_l)$ are nearly equal, under the expectation, where equality naturally holds at the sample level. $\qquad \square$

## A.2  PROOF OF LEMMA 1A AND 1 B

*Proof.* Let us rewrite the Bradley-Terry preference probability equation in terms of the DPO implicit rewards. The BT model specifies this probability of preferring $y_w$ over $y_l$ as:

$$P(y_w \succ y_l \mid x) = \sigma(r^*(x,y_w) - r^*(x,y_l)) \tag{12}$$

where the rewards can be rewritten in term of the DPO implicit rewards $\hat{r}_w$ and $\hat{r}_l$ as:

$$P(y_w \succ y_l \mid x) = \sigma\left(\underbrace{\beta \log \frac{\pi_{\theta^*}(y_w \mid x)}{\pi_{ref}(y_w \mid x)}}_{(\hat{r}_w)} - \underbrace{\beta \log \frac{\pi_{\theta^*}(y_l \mid x)}{\pi_{ref}(y_l \mid x)}}_{(\hat{r}_l)}\right) \tag{13}$$

$$= \sigma\left(\beta \log \left(\frac{\pi_{\theta^*}(y_w \mid x)\pi_{ref}(y_l \mid x)}{\pi_{\theta^*}(y_l \mid x)\pi_{ref}(y_w \mid x)}\right)\right) \tag{14}$$

$$= \sigma\left(\beta \log \left(\frac{\pi_{\theta^*}(y \mid x)\pi_{ref}(y' \mid x)}{\pi_{\theta^*}(y' \mid x)\pi_{ref}(y \mid x)}\right)\right), \quad \forall(x,y,y') \in \mathcal{D}_{nd} \subset \mathcal{D}_{pref} \tag{15}$$

where the above equation holds for all $\forall (x, y, y') \in \mathcal{D}_{\text{pref}}$, since DPO does not distinguish between the nature of the true preference relations in estimating the true preference probabilities, assuming $(y, y')$ appear as preferred and dispreferred responses respectively for any context $x$. Since this RHS of Eq. 15 is simply the sigmoided difference of implicit rewards assigned by DPO to estimate the true preference probabilities, it is straightforward to see from Proposition 1 that the RHS i.e., $\frac{\pi_{\theta*}(y)\pi_{\text{ref}}(y')}{\pi_{\theta*}(y')\pi_{\text{ref}}(y)} \to 1$ and $P(y_w \succ y_l \mid x) \sim \frac{1}{2}$, as per our definition of non-deterministic preferences. Since the reference model $\pi_{\text{ref}}$ is assumed to have full support over the output space $(\text{supp}(\pi_{\text{ref}}) = \mathcal{Y})$ and is not updated, without losing any generality, this implies that $\frac{\pi_{\theta*}(y)}{\pi_{\theta*}(y')}$ must remain close to 1 to satisfy this constraint $(\frac{\pi_{\theta*}(y)\pi_{\text{ref}}(y')}{\pi_{\theta*}(y')\pi_{\text{ref}}(y)} \to 1)$ in which case the policy tends to underfit the preference distribution since the preference signals are weak and policy cannot distinguish between the preferred and the dispreferred response.

Similar to Azar et al. (2023)'s argument, we can argue here that in this case when true preference probabilities are $\sim \frac{1}{2}$, i.e., non-deterministic, DPO's empirical reward difference estimates actually tend toward one which leads to underfitting of the optimal policy $\pi_{\theta*}$ during alignment. Indeed, in this case, the $\beta$ parameter does not provide any additional regularization effect to prevent policy underfitting especially under finite data. This completes the proof of Lemma 1a. $\qquad\square$

We can similarly prove Lemma 1b in the case when $|\mathcal{D}_{\text{nd}}| \ll N$ where $N$ is assumed to be finite. In this case, in a similar vein as Azar et al. (2023), there is more likelihood that DPO sigmoided reward difference estimates are 1, i.e., $r^*(x, y_w) - r^*(x, y_l) \to \infty$.

As such, from Eq. 12, it is straightforward to see that the term

$$\log \left( \frac{\pi_{\theta*}(y \mid x)\pi_{\text{ref}}(y' \mid x)}{\pi_{\theta*}(y' \mid x)\pi_{\text{ref}}(y \mid x)} \right) \to \infty$$

in Eq. 15. This implies that

$$\frac{\pi_{\theta*}(y \mid x)\pi_{\text{ref}}(y' \mid x)}{\pi_{\theta*}(y' \mid x)\pi_{\text{ref}}(y \mid x)} \to \infty.$$

## A.3 Proof of Lemma 1c

*Proof.* Our proof follows the argumentation in Fisch et al. (2024). Assume for now that all preference samples $(y, y') \in \mathcal{D}_{\text{pref}}$, including non-deterministic preference pairs, are mutually exclusive. Then, for the DPO objective (Eq. 3) to be minimized, each $\theta_y$ must correspond uniquely to $y$, where $\theta_y$ are the optimal parameters that minimize the DPO objective in each such disjoint preference pair. This implies that DPO objective over $\mathcal{D}_{\text{pref}}$ is convex in the set $\Lambda = \{\lambda_1, \ldots, \lambda_n\}$, where

$$\lambda_i = \beta \log \left( \frac{\pi_\theta(y_w^{(i)})\pi_{\text{ref}}(y_l^{(i)})}{\pi_\theta(y_l^{(i)})\pi_{\text{ref}}(y_w^{(i)})} \right), \quad \forall i \in \mathbb{N} \tag{16}$$

Now, consider the non-deterministic preference samples indexed by $j$, which belong to the set $\mathcal{D}_{\text{nd}} \subset \mathcal{D}_{\text{pref}}$. Let $j \in \{k+1, \ldots, N\}$ with the assumption $k \gg (N-k)$. Under the mutual exclusivity assumption, Eq. 16 must also hold true for the non-deterministic preference samples. Consequently, we can rewrite Eq. 16 as:

$$\lambda_j = \beta \log \left( \frac{\pi_\theta(y_w^{(j)} \mid x)\pi_{\text{ref}}(y_l^{(j)} \mid x)}{\pi_\theta(y_l^{(j)} \mid x)\pi_{\text{ref}}(y_w^{(j)} \mid x)} \right), \quad \forall j \in \mathcal{D}_{\text{nd}} \tag{17}$$

More specifically, for every $j$, the following holds at the limit for the DPO objective to converge:

$$\lim_{\lambda_j \to \infty} -\log\left(\sigma\left(\lambda_j\right)\right) = 0, \tag{18}$$

which implies that $\Lambda^* = \{\infty\}^N$ induces a set of global minimizers of the DPO objective that includes $\theta^*$ that are optimal for the set of non-deterministic preference samples, while inducing a parallel set of $\theta^*$ at convergence for deterministic samples.

Consequently, all global minimizers $\theta^*$ including those optimal on the non-deterministic samples must satisfy

$$\log \frac{\pi_\theta(y_w^{(j)})\pi_{\text{ref}}(y_l^{(j)})}{\pi_\theta(y_l^{(j)})\pi_{\text{ref}}(y_w^{(j)})} = \infty. \tag{19}$$

Since $0 < \pi_{\text{ref}}(y) < 1$ for all $y$, $\theta^*$ must satisfy

$$\frac{\pi_{\theta^*}(y_w^{(j)})}{\pi_{\theta^*}(y_l^{(j)})} = \infty, \tag{20}$$

implying $\pi_{\theta^*}(y_l^{(j)}) = 0$ and $\pi_{\theta^*}(y_w^{(j)}) > 0$ for all $i \in N$, given that $\pi_{\theta^*}(y_w^{(j)}) \leq 1$ for any $y_w^{(j)}$. Alternatively, let us define the complement of the aggregated representation of all the dispreferred responses $y_l^{(j)}$ i.e., $\phi(y_l)^c$, where $\phi(y_l)$ is the aggregation function. We thereby have,

$$\phi(y_l) = \{y \colon \exists j \in \mathbb{N} \text{ such that } y_l^{(j)} = y\}, \tag{21}$$

Under these conditions, it is clear that $\pi_{\theta^*}$ must assign the entire remaining probability mass to $\phi(y_l)$ as given below,

$$\pi_{\theta^*}(\mathcal{C}(y_l)) = \sum_{y \in \phi(y_l)} \pi_{\theta^*}(y) = 0 \tag{22}$$

$$\implies \pi_{\theta^*}(\phi(y_l)^c) = 1. \tag{23}$$

This completes the proof of Lemma 1c and thus Lemma 1.

$\square$

## A.4 PROOF OF LEMMA 2

*Proof.* Let us first rewrite the e-DPO objective over the preference dataset $\mathcal{D}_{\text{pref}}$ and examine how the optimal policy $\pi_\theta$ behaves upon convergence of this objective. For simplicity, we only consider the point-wise reward based distillation of the e-DPO formulation and do not consider ensembles of rewards. Additionally, note that the e-DPO objective does not require preference labels and can apply to any response pair.

$$\mathcal{L}_{\text{distill}}(r^*, \pi_\theta) = \mathbb{E}_{(x,y_1,y_2) \sim \mathcal{D}_{\text{pref}}} \left[ \left( r^*(x, y_1) - r^*(x, y_2) - \beta \log \frac{\pi_\theta(y_1 \mid x)\pi_{\text{ref}}(y_2 \mid x)}{\pi_\theta(y_2 \mid x)\pi_{\text{ref}}(y_1 \mid x)} \right)^2 \right] \tag{24}$$

$$= \mathbb{E}_{(x,y_1,y_2) \sim \rho} \left[ \left( r^*(x, y_1) - r^*(x, y_2) - \beta \log \frac{\pi_\theta(y_1 \mid x)}{\pi_\theta(y_2 \mid x)} + \beta \log \frac{\pi_{\text{ref}}(y_1 \mid x)}{\pi_{\text{ref}}(y_2 \mid x)} \right)^2 \right]. \tag{25}$$

As $\pi_\theta$ converges to the optimal policy $\pi_{\theta^*}$, the distillation objective $\mathcal{L}_{\text{distill}}$ should ideally approach zero and can be expressed as,

$$\lim_{\pi_\theta \to \pi_{\theta^*}} \mathcal{L}_{\text{distill}}(r^*, \pi_\theta) = 0$$

With some slight algebraic rearrangement and substituting the optimal policy $\pi_{\theta^*}$ for $\pi_\theta$ at convergence, we get:

$$r^*(x, y_1) - r^*(x, y_2) = \beta \log \frac{\pi_{\theta^*}(y_1 \mid x)}{\pi_{\theta^*}(y_2 \mid x)} - \beta \log \frac{\pi_{\text{ref}}(y_1 \mid x)}{\pi_{\text{ref}}(y_2 \mid x)}.$$

Since $(y_1, y_2)$ represents *any* response pairs without requiring them to be preference labels, we can substitute them with $(y, y') \in \mathcal{D}_{\text{nd}} \subset \mathcal{D}_{\text{pref}}$, where $y = y_w$ and $y' = y_l$. Without losing any generality,

we can now rewrite the above equation as,

$$r^*(x,y) - r^*(x,y') = \beta \log \frac{\pi_{\theta^*}(y \mid x)}{\pi_{\theta^*}(y' \mid x)} - \beta \log \frac{\pi_{\text{ref}}(y \mid x)}{\pi_{\text{ref}}(y' \mid x)} \tag{26}$$

$$= \beta \log \left( \frac{\pi_{\theta^*}(y \mid x)\pi_{\text{ref}}(y' \mid x)}{\pi_{\theta^*}(y' \mid x)\pi_{\text{ref}}(y \mid x)} \right) \tag{27}$$

Now, recall from our proof of Lemma 1b where we show that the RHS term of the above equation $\log \left( \frac{\pi_{\theta^*}(y|x)\pi_{\text{ref}}(y'|x)}{\pi_{\theta^*}(y'|x)\pi_{\text{ref}}(y|x)} \right) \to \infty$, which implies that $\frac{\pi_{\theta^*}(y|x)\pi_{\text{ref}}(y'|x)}{\pi_{\theta^*}(y'|x)\pi_{\text{ref}}(y|x)} \to \infty$. Indeed, when $|\mathcal{D}_{\text{nd}}| \ll N$ where $N$ is finite, unregularized scalar reward estimates of the true preference probabilities can in fact grow exceedingly large in the absence of any other regularization parameters, since $\beta$ by its own does not provide enough regularization as shown in our proof of Lemma 1a. Interestingly, similar arguments have also been made in previous works (Azar et al., 2023). The rest of this proof follows the same argumentation starting Eq. 19 assuming $0 < \pi_{\text{ref}}(y) < 1$.

This completes the proof of Lemma 2. □

## A.5 Gradient Derivation of the Focal-Softened Log-Odds Unlikelihood Loss

In this section, we derive and analyze DRDO loss gradient and offer insights into how to compared supervised alignment objectives such as DPO (Rafailov et al., 2023). Note that we do not analyze the reward distillation component here since it does not directly interact with the focal-softened contrastive log-"unlikelihood" term in training and since it is naturally convex considering its a squared term. Let us first rewrite our full DRDO loss, as:

$$\mathcal{L}_{\text{kd}}(r^*, \pi_\theta) = \mathbb{E}_{(x,y_1,y_2)\sim\mathcal{D}_{\text{pref}}}\left[ \underbrace{\left( r^*(x,y_1) - r^*(x,y_2) - (\hat{r}_1 - \hat{r}_2) \right)^2}_{\text{Reward Difference}} \right.$$
$$\left. - \underbrace{\alpha(1-p_w)^\gamma \log \left( \frac{\pi_\theta(y_w \mid x)}{1 - \pi_\theta(y_l \mid x)} \right)}_{\text{Contrastive Log-"unlikelihood"}} \right], \tag{28}$$

where $p_w = \sigma(z_w - z_l) = \frac{1}{1+e^{-(z_w-z_l)}}$ and quantifies the student policy's confidence in correctly assigning the preference from $z_w = \log \pi_\theta(y_w \mid x)$ and $z_l = \log \pi_\theta(y_l \mid x)$, or the log-probabilities of the winning and losing responses, respectively.

Without the expectation, consider only the focal-softened log-odds unlikelihood loss given by:

$$-\alpha \cdot (1-p_w)^\gamma \cdot \log \left( \frac{\pi_\theta(y_w \mid x)}{1 - \pi_\theta(y_l \mid x)} \right), \tag{29}$$

Taking the gradient of this term with respect to the model parameters $\theta$ and using $\sigma'(x) = \sigma(x)(1 - \sigma(x))$, we derive:

$$\nabla_\theta \mathcal{L}_{\text{kd}} = \alpha\gamma(1-p_w)^{\gamma-1}p_w(1-p_w)(\nabla_\theta z_w - \nabla_\theta z_l) \cdot \log \left( \frac{\pi_\theta(y_w \mid x)}{1 - \pi_\theta(y_l \mid x)} \right) \tag{30}$$

$$- \alpha(1-p_w)^\gamma \left( \frac{\nabla_\theta \pi_\theta(y_w \mid x)}{\pi_\theta(y_w \mid x)} + \frac{\nabla_\theta \pi_\theta(y_l \mid x)}{1 - \pi_\theta(y_l \mid x)} \right). \tag{31}$$

$$\nabla_\theta \mathcal{L}_{\mathrm{kd}} = \alpha\gamma(1-p_w)^\gamma p_w(\nabla_\theta z_w - \nabla_\theta z_l) \cdot \log\left(\frac{\pi_\theta(y_w \mid x)}{1 - \pi_\theta(y_l \mid x)}\right) \tag{32}$$

$$-\alpha(1-p_w)^\gamma \left(\underbrace{\frac{\nabla_\theta \pi_\theta(y_w \mid x)}{\pi_\theta(y_w \mid x)}}_{\text{increase } \pi_\theta(y_w|x)} + \underbrace{\frac{\nabla_\theta \pi_\theta(y_l \mid x)}{1 - \pi_\theta(y_l \mid x)}}_{\text{decrease } \pi_\theta(y_l|x)}\right). \tag{33}$$

While this above equation might appear rather cumbersome, notice that in preference learning in language models, the output token space $\mathcal{Y}$ is exponentially large. Additionally, in typical bandit settings, we consider the entire response itself as the action (summation of log probabilities). Since the modulating term $0 \le (1-p_w)^\gamma \le 1$ and $\alpha$ is typically small $\sim 0.1$ (Lin et al., 2018; Yi et al., 2020) compared to gradients appearing in likelihood terms appearing above, we can conveniently ignore the first term for the gradient analysis.

Simplifying the above equation, we get

$$-\alpha(1-p_w)^\gamma \left(\underbrace{\frac{\nabla_\theta \pi_\theta(y_w \mid x)}{\pi_\theta(y_w \mid x)}}_{\text{increase } \pi_\theta(y_w|x)} + \underbrace{\frac{\nabla_\theta \pi_\theta(y_l \mid x)}{1 - \pi_\theta(y_l \mid x)}}_{\text{decrease } \pi_\theta(y_l|x)}\right). \tag{34}$$

We can now draw some insights and direct comparisons of our approach with Direct Preference Optimization (DPO) (Rafailov et al., 2023). As in most contrastive preference learning gradient terms (Rafailov et al., 2023; Hong et al., 2024; Xu et al., 2024; Meng et al., 2024; Ethayarajh et al., 2024), the term $\frac{\nabla_\theta \pi_\theta(y_w|x)}{\pi_\theta(y_w|x)}$ in Eq. 34 amplifies the gradient when $\pi_\theta(y_w \mid x)$ is low, driving up the likelihood of the preferred response $y_w$. Similarly, $\frac{\nabla_\theta \pi_\theta(y_l|x)}{1-\pi_\theta(y_l|x)}$ penalizes overconfidence in incorrect completions $y_l$ when $p_w$ is low, encouraging the model to hike preferred response likelihood while discouraging dispreferred ones.

The key insight here is that the modulating term, $(1-p_w)^\gamma$, strategically amplifies corrections for difficult examples where the probability $p_w$ of the correct (winning) response is low. Intuitively, unlike DPO's fixed $\beta$ that is applied across the whole training dataset, this modulating term amplifies gradient updates when preference signals are weak ($p_w \approx 0.5$) and tempering updates when they are strong ($p_w \approx 1$), thus ensuring robust learning across varying preference scenarios. Intuitively, when ($p_w \approx 1$), the model is already confident of its decision since $p_w$ remains high, indicating increased model confidence for deterministic preferences. In contrast, when $p_w$ is small, $(1-p_w)$ remains near 1, and the term $(1-p_w)^\gamma$ retains significant magnitude, especially for larger values of $\gamma$. This allows $\pi_\theta$ in DRDO to learn from both deterministic and non-deterministic preferences, effectively blending reward alignment with preference signals to guide optimization.

For a deeper intuition, consider the case where the true preference probabilities from $p^*(x,y) \sim \frac{1}{2}$. In this case, since non-deterministic preference samples are typically low, from Proposition 1, DPO would assign zero difference in its implicit rewards, especially for finite preference data. Then as Lemma 1(a) suggests, DPO gradients would effectively be nullified, regardless of $\beta$ since the its reward difference range $\in (-\infty, +\infty)$. In this case as $\pi_\theta$ cannot distinguish between the preference pair and, in turn, effectively misses out on the preference information for such samples.

On the other hand, for $|\mathcal{D}_{\mathrm{nd}}| \ll N$, if $\pi_\theta$ in DPO estimates the true preference close to 1 where $\hat{p} = 1$, as Lemma 1(b) suggests, the empirical policy would assign very high probabilities to tokens that do not even appear in the data. This leads to a surprising combination of both underfitting and overfitting, except the overfitting here results in DPO policy generating tokens that are irrelevant to the context.

However, as Lemma 2 suggests, e-DPO does not directly face this limitation *during* training, but the degeneracy manifests upon convergence. Under the same conditions, the DRDO loss still operates at a sample level because if the DRDO estimate of $p^*$ (via $p_w$) is close to its true value of $\sim \frac{1}{2}$, the modulating factor ensures that gradients do not vanish. This allows DRDO to continue learning from such samples until convergence when winning and losing probabilities are pushed further apart

(Fig. 2). Intuitively, the $\log\left(\frac{\pi_\theta(y_w|x)}{1-\pi_\theta(y_l|x)}\right)$ term is minimized precisely under this condition where the modulating term is also close to zero.

---

**DRDO Algorithm**

**Input:** Preference dataset $\mathcal{D}_{\text{pref}} = \{(x^{(i)}, y_w^{(i)}, y_l^{(i)})\}_{i=1}^N$, initialized policy model with reward head $\pi_{\theta,\theta'} \leftarrow \text{SFT}(\theta) \oplus r_{\theta'}$.
**Output:** Optimized model parameters $\theta$ in policy $\pi_\theta$.

1. Train Oracle $r_\phi$ with loss $\mathcal{L}_\mathcal{O}(r_\phi, \mathcal{D}_{\text{pref}})$ (see Eq. 5).
2. For $t = 1, \dots, T$:
   (a) For each $(x^{(i)}, y_w^{(i)}, y_l^{(i)})$ in $\mathcal{D}_{\text{pref}}$:
       i. Compute $r_1^* = r_\phi(x^{(i)}, y_w^{(i)})$ and $r_2^* = r_\phi(x^{(i)}, y_l^{(i)})$.
       ii. Compute $\hat{r}_1 = r_{\theta'}(x^{(i)}, y_w^{(i)})$ and $\hat{r}_2 = r_{\theta'}(x^{(i)}, y_l^{(i)})$.
       iii. Compute knowledge distillation loss:

$$\mathcal{L}_{\text{kd}}(r^*, \pi_\theta) = \mathbb{E}_{(x^{(i)}, y_w^{(i)}, y_l^{(i)}) \sim \mathcal{D}_{\text{pref}}}\left[ \underbrace{\left(r_1^* - r_2^* - (\hat{r}_1 - \hat{r}_2)\right)^2}_{\text{Reward Difference}} - \underbrace{\alpha(1-p_w)^\gamma \log\left(\frac{\pi_\theta(y_w^{(i)} \mid x^{(i)})}{1-\pi_\theta(y_l^{(i)} \mid x^{(i)})}\right)}_{\text{Contrastive Log-"unlikelihood"}} \right].$$

       iv. Update $\pi_{\theta,\theta'}$ using $\mathcal{L}_{\text{kd}}$.
3. **Return:** Aligned policy $\pi_\theta$.

---

Table 5: DRDO Algorithm steps. We start off with the preference dataset and an SFT-trained policy initialized with an additional linear head parameterized by $\theta'$. Once our oracle is trained, we compute both estimated rewards for each response ($y$) from the initial policy ($\hat{r}$) as well as from the oracle ($r^*$). We then use $\mathcal{L}_{\text{kd}}$ to update both $\theta$ and $\theta'$ in $\pi_\theta$ resulting in our DRDO aligned policies.

## B  FURTHER NOTES ON EXPERIMENTAL SETUP

We provide the following additional explanatory notes regarding the experimental setup:

- For non-deterministic and nuanced preferences, note that although we fine-tune all approaches (including DRDO) on `https://huggingface.co/datasets/CarperAI/openai_summarize_tldr`, every baseline we use is policy-aligned with only the training data within $\mathcal{D}_{all}$, $\mathcal{D}_{hc,he}$ and $\mathcal{D}_{\ell c,\ell e}$ for a direct comparison.

- Pal et al. (2024) assume non-determinism of preferences to be correlated to edit-distances between pairwise-samples, but we do not make sure assumptions and consider both the true (oracle) rewards and edit-distances between pairs to verify the robustness of our method. Pal et al. (2024)'s theoretical framework brings insights on DPO's suboptimality assumes small edit distance between pairwise samples and they empirically show this primarily for math and reasoning based tasks. In contrast, our evaluation framework is more general in the sense that we consider both the oracle reward difference as well as edit distance in addressing DPO's limitations in learning from non-deterministic preferences and we evaluate on a more diverse set of prompts apart from math and reasoning tasks.

- For reward distillation w.r.t to model size, the version of Ultrafeedback we used can be found at `https://huggingface.co/datasets/argilla/ultrafeedback-binarized-preferences-cleaned`.

- For SFT on both experiments, use the TRL library implementation (`https://huggingface.co/docs/trl/en/sft_trainer`) for SFT training on all initial policies for all baselines.

### B.1  CHOICE OF EVALUATION DATASETS

As mentioned in Sec. 5, our experiments were designed to balance robustness and thoroughness with research budget constraints, and to account for properties of the task being evaluated relative to the data. The rationale for evaluating DRDO's robustness to non-deterministic or ambiguous preferences

is straightforward: the TL;DR dataset is annotated with human confidence labels, which enables the creation of the $\mathcal{D}_{hc,he}$ and $\mathcal{D}_{\ell c,\ell e}$ splits requires to test the different non-determinism settings.

Testing the effect of model size on reward distillation does not require human confidence labels, and as the TL;DR training data size is 1.3M rows, testing distillation across 5 models became computationally intractable given available resources. Thus we turned to Ultrafeedback, which at a train size of 61.1k rows made this experiment much more feasible. Additionally, Ultrafeedback is rather better suited to the preference distillation problem because Ultrafeedback was specifically annotated and cleaned (Cui et al., 2024), for the purposes of evaluating open source models for distillation. Ultrafeedback also provides a high quality dataset with GPT-4 or scores on both straighforward summarization instruction following *and* multiple preference dimensions such as honesty, truthfulness, and helpfulness. Previous work like Zephyr (Tunstall et al., 2023) utilize this dataset for evaluating preference distillation. However, their method is more of a data-augmentation method and does not provide any novel distillation algorithms for preferences since they use the DPO loss but with cleaner and diverse feedback data. They only test student models of ∼7B parameters, which are still relatively large models that require additional compute for training without including some form of PEFT. As such, Ultrafeedback is best suited to evaluate distillation-based methods and our novel contribution in this area included evaluation of smaller distilled models.

## B.2 TL;DR SUMMARIZATION DATASET SPLITS

Table 6: Mean confidence and normalized edit distance statistics our TL;DR preference generalization experiment.

| Split | Conf (Mean) | Edit Dist (Mean) | Train | Validation |
|---|---|---|---|---|
| $\mathcal{D}_{all}$ | 5.01 | 0.12 | 44,709 | 86,086 |
| $\mathcal{D}_{hc,he}$ | 7.31 | 0.15 | 10,136 | 1,127 |
| $\mathcal{D}_{\ell c,\ell e}$ | 2.67 | 0.09 | 10,000 | 1,112 |

Table 6 shows the mean confidence and normalized edit distance statistics in the TL;DR dataset which we used to compute the deterministic and non-determinisitic splits. $\mathcal{D}_{all}$ represents the full data (Stiennon et al., 2020). $\mathcal{D}_{hc,he}$ and $\mathcal{D}_{\ell c,\ell e}$ represent subsets created by splitting at the 50th percentile of human labeler confidence and edit distance values. The range of labeler confidence values for the full training data range is $[1, 9]$. Additionally, segmenting the training data based on the *combination* of confidence and edit distance thresholds do not make the splits roughly half of the full training data. This is because there are many samples that do not simultaneously satisfy the 50th percentile threshold for each metric. In reality, this choice is intentional to more robustly evaluate DRDO under more difficult preference data settings. Additionally, previous theoretical as well as empirical work (Pal et al., 2024) has shown that supervised methods like DPO fail to learn optimal policies when the token-level similarity is high in the preference pairs, especially in the beginning of the response. Therefore, we apply this combined thresholding for all our experiments on TL;DR summarization dataset.

## C PREFERENCE LIKELIHOOD ANALYSIS

Fig. 4 shows preference optimization method performance vs. training steps, according to Bradley-Terry (BT) implicit reward accuracies (Fig. 4a), Oracle reward advantage (Fig. 4b), preferred log-probabilities (Fig. 4c) and dispreferred log-probabilities (Fig. 4d). Although all baselines show roughly equal performance in increasing the likelihood of preferred responses ($y_w$), DRDO is particularly efficient in penalizing dispreferred responses ($y_l$) as Fig. 4d suggests.

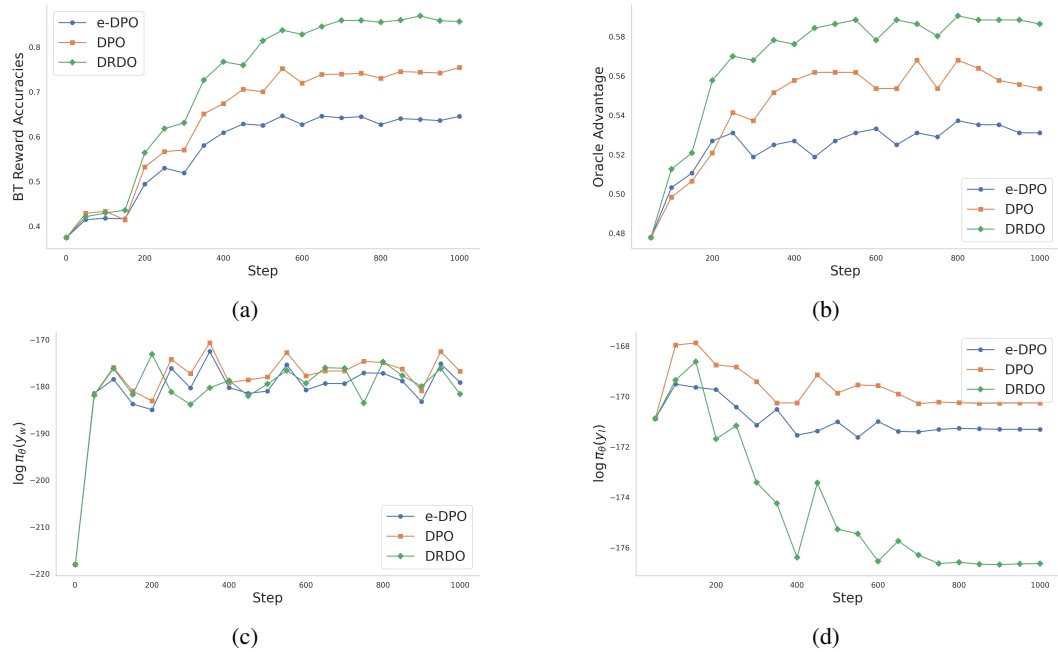

Figure 4: Top: DRDO performance evolution during OPT 1.3B training compared to DPO and e-DPO on the evaluation set of Ultrafeedback (Cui et al., 2024), and randomly sampled generations to compute the reward advantage against the preferred reference generations.

# D HYPERPARAMETERS

Table 7: Model Configuration and Full set of hyperparameter used for DRDO Training

| Parameter | Default Value |
|---|---|
| learning_rate | 5e-6 |
| lr_scheduler_type | cosine |
| weight_decay | 0.05 |
| optimizer_type | paged_adamw_32bit |
| loss_type | DRDO |
| per_device_train_batch_size | 12 |
| per_device_eval_batch_size | 12 |
| gradient_accumulation_steps | 4 |
| gradient_checkpointing | True |
| gradient_checkpointing_use_reentrant | False |
| max_prompt_length | 512 |
| max_length | 1024 |
| max_new_tokens | 256 |
| max_steps | 20 |
| logging_steps | 5 |
| save_steps | 200 |
| save_strategy | no |
| eval_steps | 5 |
| log_freq | 1 |
| $\alpha$ | 0.1 |
| $\gamma$ | 2 |

We only use full-parameter training for all our policy models. We train all our student policies for 1k steps with an effective batch size of 64, after applying an gradient accumulation step of 4.

Specifically, both OPT series and Phi-3-Mini-4K-Instruct model were optimized using DeepSpeed ZeRO 2 (Rasley et al., 2020) for faster training. All models were trained on 2 NVIDIA A100 GPUs, except for certain runs that were conducted on an additional L40 gpu. For the optimizer, we used AdamW (Loshchilov et al., 2017) and paged AdamW (Dettmers et al., 2024) optimizers with learning rates that were linearly warmed up with a cosine-scheduled decay. For both datasets, we filter for prompt and response pairs that are $< 1024$ tokens after tokenization. This allows the policies enough context for coherent generation. Apart from keeping compute requirement reasonable, this avoids degeneration during inference since we force the model to only generate upto 256 new tokens not including the prompt length in the maximum token length.

For our DRDO approach, we sweep over $\alpha \in \{.1, 1\}$ and $\gamma \in \{0, 1, 2, 5\}$ but found the most optimal combination to be $\alpha = 0.1$ and $\gamma = 2$, since a higher $\gamma$ tends to destabilize training due to the larger penalties induced on DRDO loss. This is consistent with optimal $\gamma$ values found in the literature, albeit for different tasks (Yi et al., 2020; Lin et al., 2018). For Oracle trained for DRDO, we use the same batch size as for policy training with a slightly larger learning rate of 1e-5 and train for epoch. For consistency, we use a maximum length of 1024 tokens after filtering for pairs with prompt and responses < 1024 tokens. For all SFT training, we use the TRL library[8] with a learning rate of 1e-5 with a cosine scheduler and 100 warmup steps.

For the DPO baselines, we found the implementation in DPO Trainer[9] For optimal parameter selection, we sweep over $\beta \in \{.1, 0.5, 1, 10\}$ but we found the default value of $\beta = 0.1$ to be most optimal based on the validation sets during training. Also, we found the default learning rate of 5e-6 to be optimal after validation runs. For e-DPO, we restrict the number of reward ensembles to 3 but use the same Oracle training hyperparameters mentioned above.

Table 7 provides a full list of model configurations and hyperparameters used during trainng of DRDO models.

# E    WIN-RATE EVALUATION PROMPT FORMATS

Fig. 6 and Fig. 7 show the prompt format used for GPT-4o evaluation of policy generations compared to human summaries provided in the evaluation data of Reddit TL;DR (CNN Daily Articles). Fig. 6 specifically provides the human-written summaries as reference in GPT's evaluation of the baselines. In contrast, Fig. 7 shows the prompt that was used to evaluate policy generated summaries in direct comparison to human-written summaries. Note that in both prompts, we swap order of provided summaries to avoid any positional bias in GPT-4o's automatic evaluation.

## E.1    FURTHER ABLATIONS USING GPT-4O AS ORACLE

For a more robust evaluation using a much more high-capacity oracle (GPT-4o), we randomly sampled 40 prompts from the CNN daily test set and compute win-rates and reward margins of samples generated with a top-p of 0.8 and T=0.7 for all baselines vs. DRDO policies trained on Reddit TL;DR. We use the Phi-3-Mini-4K-Instruct model for this experiment. We include the IPO baseline (Azar et al., 2024) with $\beta = 0.1$ (or $\tau$ in their paper), the baselines without the distillation (shown as DRDO (-R)) and contrastive component (shown as DRDO (-C)). Additionally, we include the baseline where reward distillation term in DRDO is replaced by the DPO loss but keep the contrastive log-unlikelihood component (shown as DPO (-C)). Tab. 8 shows results of this experiment. Note that due to compute constraints, we only train these policies from the SFT checkpoint for 500 steps with an effective batch size of 128. For the reward estimates using GPT-4o, we only add one additional condition to the prompt shown in Fig. 5 to get scalar rewards (between 0 and 1). Fig. 8 provides the prompt format used for this experiment.

## E.2    DRDO PERFORMANCE VS EXTENT OF OUT-OF-DISTRIBUTION DATA

In order to comprehensively evaluate DRDO's performance against baseline methods under increasing out-of-distribution (OOD) conditions, we randomly sample 1,000 prompts from the CNN/DailyMail

---

[8]https://huggingface.co/docs/trl/en/sft_trainer
[9]https://huggingface.co/docs/trl/main/en/dpo_trainer to be the most stable and build off most of our DRDO training pipline and configuration files based on their trainer.

| Comparison | WR A (%) | WR B (%) | Reward A | Reward B | Margin A | Margin B |
|---|---|---|---|---|---|---|
| DRDO vs. DRDO (-R) | 85.0 | 15.0 | $0.24_{\pm 0.26}$ | $0.07_{\pm 0.08}$ | $0.22_{\pm 0.22}$ | $0.09_{\pm 0.04}$ |
| DRDO vs. DRDO (-C) | 90.0 | 10.0 | $0.25_{\pm 0.23}$ | $0.05_{\pm 0.07}$ | $0.23_{\pm 0.22}$ | $0.06_{\pm 0.03}$ |
| DRDO vs. IPO | 65.0 | 35.0 | $0.21_{\pm 0.17}$ | $0.21_{\pm 0.24}$ | $0.09_{\pm 0.08}$ | $0.16_{\pm 0.16}$ |
| DRDO vs. DPO (-C) | 80.0 | 20.0 | $0.18_{\pm 0.16}$ | $0.14_{\pm 0.15}$ | $0.11_{\pm 0.08}$ | $0.22_{\pm 0.21}$ |

Table 8: DRDO policies (shown as A) compared against various baselines (shown as B)–win-rates (WR) are computed using average of reward comparison for each sample and then averaged. Margins are computed using the difference of rewards.

dataset and segment these prompts into bins of 50 tokens based on prompt-token counts, spanning the full range of token lengths. Since the CNN/DailyMail dataset represents a previously unseen input distribution, this evaluation effectively measures the OOD generalization capabilities of the policies as prompt lengths (and corresponding news article lengths) increase. For this automatic evaluation, we use GPT-4o as a high-capacity judge, consistent with Sec. 5 (prompt used is shown in Fig. 6). For response sampling, we use top-p of 0.8 and temperature of 0.7 for DRDO and all baselines. The trends in Fig. 5 suggest that as OOD composition (with prompt-token lengths as proxy) increases, on average DRDO policies tend to have relatively larger win-rates compared to shorter prompts over baselines like DPO and e-DPO. In contrast, we find that DPO and e-DPO win-rates tend to decrease with increase in prompt-lengths.

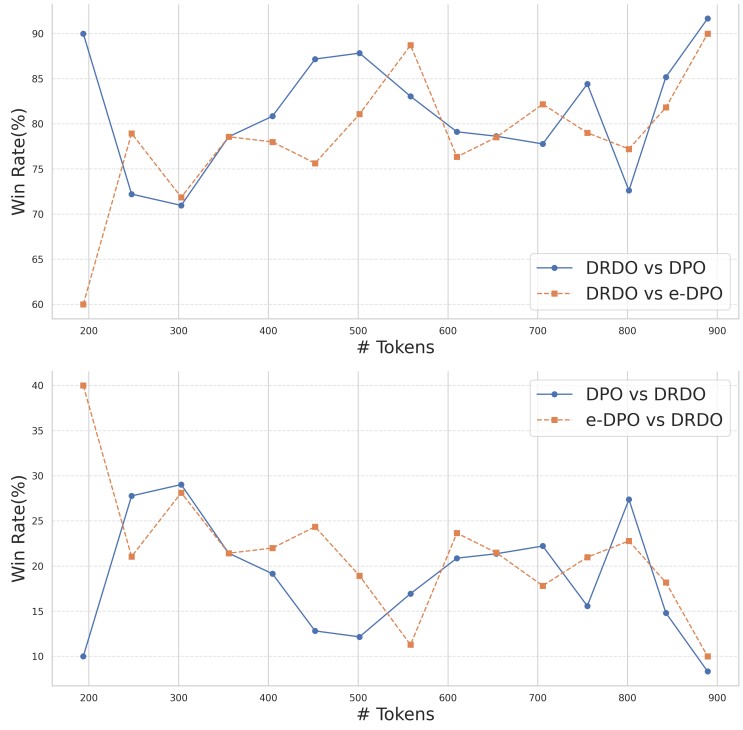

Figure 5: Comparison of win-rates as a function of the extent of out-of-distribution (OOD) data on the CNN daily article dataset. Win-rates (y-axis) of DRDO vs DPO and e-DPO (top) and competitor win-rates (bottom) are plotted against the increasing prompt lengths (number of tokens) over 1000 randomly sampled prompts for evaluation. DRDO is more robust to OOD settings, on average, compared to baselines like DPO and e-DPO as seen in the upward trend in win-rates over prompt-tokens.

## F COMPUTATIONAL EFFICIENCY

e-DPO requires training reward ensembles to form a confidence set for training the policy. In our experiments, we use 3 reward models to construct this set which makes it roughly thrice as expensive as DRDO training. Like DPO, e-DPO requires a separate reference model to be kept in memory,

**Summarization GPT-4 win rate prompt (C).**

```
Which of the following summaries do a better job of summarizing the most \
important points in the given forum post, without including unimportant \
or irrelevant details? Make your decision while referring to the reference\
(human-written) summary. A good summary is both precise and concise.

Post:
<post>

Reference Summary:
<golden summary>

Summary A:
<Summary A>
Summary B:
<Summary B>

FIRST provide a one-sentence comparison of the two summaries, explaining \
which you prefer and why. SECOND, on a new line, state only "A" or "B" to \
indicate your choice. Your response should use the format:
Comparison: <one-sentence comparison and explanation>
Preferred: <"A" or "B">
```

Figure 6: Prompt format for Reddit TL;DR (CNN Daily)

**Summarization GPT-4 win rate prompt (vs human written).**

```
Which of the following summaries does a better job of summarizing the most \
important points in the given news article, without including unimportant \
or irrelevant details? A good summary is both precise and concise.

Post:
<post>

Summary A:
<golden summary>
Summary B:
<summary>

FIRST provide a one-sentence comparison of the two summaries, explaining \
which you prefer and why. SECOND, on a new line, state only "A" or "B" to \
indicate your choice. Your response should use the format:
Comparison: <one-sentence comparison and explanation>
Preferred: <"A" or "B">
```

Figure 7: Prompt format for Reddit TL;DR (CNN Daily)

further increasing compute requirements. DRDO, on the other hand, only requires a trained oracle for distillation. The expected oracle rewards can be precomputed once and a separate reference model does not need to be kept in memory (as shown in Eq. 6). During training, DRDO does require one additional linear head on top of the base LM to predict the reward estimates. This adds a negligible 0.003% more trainable parameters (relative to the language modeling head of base LM Phi-3-Mini-4K-Instruct). During inference, DRDO trained policies do not require this head.

## G   DRDO VS PLURALISTIC PREFERENCES

In certain circumstances, non-deterministic preferences, as reflected in low labeler confidence or equal rewards, could be a consequence of innate pluralistic tendencies of human preferences. However, DRDO is not motivated directly by pluralistic preferences, where there are multiple annotations (or preferences) for a single $(x, y_1, y_2)$, but by the diversity of preference strength for paired samples.

---

**Summarization GPT-4o win rate prompt (C).**

Which of the following summaries do a better job of summarizing the most important points in the given forum post, without including unimportant or irrelevant details? Make your decision while referring to the reference (human-written) summary. A good summary is both precise and concise.

**Post:**
```
<post>
```

**Reference Summary:**
```
<golden summary>
```

**Summary A:**
```
<Summary A>
```
**Summary B:**
```
<Summary B>
```

FIRST, provide a one-sentence comparison of the two summaries, explaining which you prefer and why.
SECOND, on a new line, state only **"A"** or **"B"** to
indicate your choice. Your response should use the format:
THIRD, on a new line, provide your ratings (a real reward score between 0 to 1 where 1 is highest and 0 is lowest in quality) for the summaries.
```
Comparison:  <one-sentence comparison and explanation>
Preferred:  <"A" or "B">
Score for Summary A: <score>
Score for Summary B: <score>
```

---

Figure 8: Prompt format for Reddit TL;DR

Typically, pluralistic approaches require multiple reward models (rewarded soups Ramé et al. (2023), e-DPO Fisch et al. (2024), MaxMin-RLHF Chakraborty et al., or conditioned policy Wang et al. (2024b)) to model such preferences. This is computationally expensive and *assumes rewards over multiple dimensions can be linearly interpolated*. We do not make any such assumptions: 1) our main argument is that non-deterministic preferences likely constitute a non-trivial amount of paired samples in popular preference datasets and as such, DRDO provides an efficient alignment method under such conditions. 2) our only strong assumption in the modeling is that the Oracle reward model, given sufficient data, should reasonably approximate human preferences using any standard reward-modeling approach. Furthermore, given such an oracle, we directly regress on the rewards and, unlike e-DPO, do not need to find additional optimal parameters like $\beta$ in the regression or confidence set in policies or reward model ensembles. Thus our DRDO approach does not need to learn a variety of models each unique to specific viewpoints expressed in the data, and thus our results that best the competitor baselines reflect that we are able to fit better to non-deterministic preferences while still maintaining an ability to fit to deterministic preferences and the data distribution at large.

## H SENSITIVITY OF DRDO'S $\gamma$ VS DPO'S $\beta$ W.R.T KL-DIVERGENCE OVER SFT MODEL

We ran an additional experiment to compare the sensitivity of model-specific hyperparameters (DRDO's $\gamma$ vs DPO's $\beta$). Keeping $\alpha$ as 0.1 for all DRDO policies, we compute the KL-divergence during training on sampled generations on 40 randomly sampled evaluation prompts in the held-out set of Ultrafeedback with top-p of 0.8 and temperature of 0.7 with various $\gamma$ values in DRDO ($\alpha = 0.1$) and with different KL-$\beta$ values in DPO using the SFT-trained Phi-3-Mini-4K-Instruct model. Tab. 9 shows expected KL-divergence (averaged over tokens) over the 40 completions at every 100 steps of training. The expected reward accuracies (win-rates) over the SFT model completions with the same hyperparameters over these samples (after 400 training steps) are shown in Tab. 10 below.

These results suggest that while DRDO does not explicitly regularize its policy wrt reference-model based KL regularization, it still outperforms DPO in oracle-assigned expected reward accuracies (win-rates) on sampled generations as long as the $\gamma$ parameter is carefully chosen. In particular, as previously observed in Meng et al. (2024); Rafailov et al. (2024) , smaller $\beta$ in DPO tends to increase KL-divergence with respect to the baseline SFT model. However, a relatively larger KL divergence in DRDO on average does not necessarily impede preference learning but larger $\gamma$ values tend to degrade expected rewards.

| Step | DRDO ($\gamma = 5$) | DRDO ($\gamma = 2$) | DRDO ($\gamma = 1$) | DPO ($\beta = 0.01$) | DPO ($\beta = 0.1$) |
|------|------|------|------|------|------|
| 100 | 0.63 | 0.44 | 0.82 | 0.64 | 0.38 |
| 200 | 0.35 | 1.51 | 1.67 | 0.81 | 0.39 |
| 300 | 0.59 | 1.67 | 1.82 | 1.17 | 0.42 |
| 400 | 0.64 | 1.63 | 1.71 | 1.35 | 0.44 |

Table 9: KL-divergence during training on sampled generations on 40 randomly sampled evaluation prompts in the held-out set of Ultrafeedback with top-p of 0.8 and temperature of 0.7 with various $\gamma$ values in DRDO ($\alpha = 0.1$) and with different KL-$\beta$ values in DPO using the Phi-3-Mini-4K-Instruct model.

| Model | Expected Oracle Reward |
|-------|------------------------|
| DPO ($\beta = 0.1$) | 0.775 ($\pm$ 0.42) |
| DPO ($\beta = 0.01$) | 0.675 ($\pm$ 0.47) |
| DRDO ($\gamma = 1$) | 0.750 ($\pm$ 0.44) |
| DRDO ($\gamma = 2$) | 0.825 ($\pm$ 0.38) |
| DRDO ($\gamma = 5$) | 0.600 ($\pm$ 0.50) |

Table 10: DRDO vs DPO expected reward accuracies (win-rates) over the SFT-model completions computed using the OPT 1.3B oracle model.

As for the exponential parameter $\gamma$, $\gamma = 2$ appears to be a reasonable choice, as previously found in in Lin et al. (2018); Yi et al. (2020) in the focal loss literature. A larger $\gamma$ can harshly penalize the loss when the policy is uncertain ($p_w << 1$) while a smaller $\gamma = 0$ may not adequately penalize and impact its adaptive nature. In our experiments including the above experiment, we find that the optimal reward is achieved for $\gamma = 2$ while too low or too high a $\gamma$ can affect performance as seen in Tab. 10 . Note that, although we find $\gamma = 2$ to be optimal across datasets, a reasonable way to find the right $\gamma$ would vary case by case–if the baseline policy at the start of alignment training has not undergone or in off-policy settings, a lower $\gamma$ could be ideal since a higher $\gamma$ might apply harsher penalties in this case. However, if the policy is initialized with SFT model (as in DRDO) or in on-policy (where $p_w$ is likely to be higher already) alignment settings, a higher $\gamma$ could be optimal. In practice though, empirical validation on a held-out set can be an efficient alternative, similar to how an optimal $\beta$ can be determined in algorithms like DPO.

