# OpenReview forum: "Simultaneous Reward Distillation and Preference Learning: Get You a Language Model Who Can Do Both"
_ICLR.cc/2025/Conference — Submitted to ICLR 2025_

### Official Review · Reviewer_G9Er · 2024-10-22

**Soundness:** 1
**Presentation:** 3
**Contribution:** 2
**Rating:** 3
**Confidence:** 4

**Summary:**

This work seeks to addresses challenges that arise in DPO when ‘non-deterministic’ preferences are present in the dataset. The authors provide a theoretical motivation behind why these points are challenging and propose DRDO, Direct Reward Distillation and policy-Optimization to address this issue. They test DRDO on subsets of the Tldr and Ultrafeedback datasets.

**Strengths:**

The proposal of the paper to combine a reward difference loss that measures the strength of preferences with a contrastive loss term that uses the binary preference labels from preference datasets is novel to the best of my knowledge. The authors show the proposed approach produces an improved win rate over a DPO and e-DPO baseline.

The paper conducts varied experiments to understand the proposed method. They vary the size and temperature of the generated responses to explore a wide range of possible settings and analyse the win rate using suitable approaches and prompts with GPT 4.0.

**Weaknesses:**

A key component of the paper is the theoretical analysis of the weaknesses of DPO, specifically Lemma 1 and Lemma 2. The proof of these Lemma in the appendices seems to have multiple mistakes, for example the statement at the end of Appendix A.2 appears wrong as the latter equation should approach 1, this issue is repeated in the derivation of Lemma 2, both are key for later motivating the proposed DRDO. If this is a misunderstanding on my behalf, the quality of the explanation needs improvement as both Section 3 and the writing in the appendix was hard to parse.

Furthermore, it appears that Lemma 1 simply re-states results found within the existing literature, issues with deterministic preferences in DPO i.e. Lemma 1a) and 1b) are presented in [1] and the result of Lemma 1c) appear in [2], both works are cited by this paper.
The motivation of the paper is also very unclear to me. The authors define noisy preferences as those with $p(y \succ y’) \approx ½$ and claim that “This (DPO) formulation is challenged by non-deterministic or noisy preference labels”. In the proposed setting it appears that these non-deterministic points are just weak learning signals. The authors should justify why these points matter, for example, pluralistic approaches argue that these weak signals stem from multiple different viewpoints being present in the dataset and thus work to learn models unique to specific viewpoints [3,4]. An argument that motivates the proposed approach against the pluralistic setting would be beneficial.

[1] Mohammad Gheshlaghi Azar, Mark Rowland, Bilal Piot, Daniel Guo, Daniele Calandriello, Michal Valko, and Rémi Munos. A general theoretical paradigm to understand learning from human preferences, 2023.

[2] Adam Fisch, Jacob Eisenstein, Vicky Zayats, Alekh Agarwal, Ahmad Beirami, Chirag Nagpal, Pete Shaw, and Jonathan Berant. Robust preference optimization through reward model distillation, 2024. URL https://arxiv.org/abs/2405.19316.

[3] Souradip Chakraborty, Jiahao Qiu, Hui Yuan, Alec Koppel, Dinesh Manocha, Furong Huang, Amrit Bedi, and Mengdi Wang. 2024. MaxMin-RLHF: Alignment with Diverse Human Preferences. In ICML. https://proceedings.mlr.press/v235/chakraborty24b.html

[4] Kaiwen Wang, Rahul Kidambi, Ryan Sullivan, Alekh Agarwal, Christoph Dann, Andrea Michi, Marco Gelmi, Yunxuan Li, Raghav Gupta, Avinava Dubey, et al. Conditioned language policy: A general framework for steerable multi-objective finetuning. arXiv preprint arXiv:2407.15762, 2024.

**Questions:**

1.	In Figure 2, is the DRDO Preference Loss $L_{kl}$ or the Contrastive Log-unlikelihood?

2.	What does an ablation on the two loss terms look like? If trained just with the contrastive loss how does the model perform?

3.	Whilst the paper compares to e-DPO how does the approach compare to IPO an alternative to DPO mentioned heavily in the first few sections of the paper that does not have a reward oracle component?

---

> ### Author Response · Authors · 2024-11-21
> **Correction in lemma**
>
> Thanks for your detailed comments. Our responses are below and revisions to the main paper have been made in red in the revised PDF.
>
> Typos in Lemma Proofs:
> Thank you for catching the issue! The mistakes in the proof stem from typographical errors and do not affect the main conclusions of Lemma 1. In Lemma 1a, the ratio of likelihoods (without the log) should tend to 1 instead of 0. More specifically, since the reference model \\( \pi_{\mathrm{ref}} \\) is assumed to have full support over the output space (\\( \text{supp}(\pi_{\text{ref}}) = \mathcal{Y} \\)) and is *not* updated, without losing any generality, this implies that \\( \frac{\pi_{\theta^*}(y)}{\pi_{\theta^*}(y')} \\) must remain close to 1 to satisfy this constraint:
> \\[
> \frac{\pi_{\theta^*}(y)\pi_{\mathrm{ref}}(y')}{\pi_{\theta^*}(y')\pi_{\mathrm{ref}}(y)} \to 1.
> \\]
> In this case, the policy still (after correcting the mistake) tends to underfit the preference distribution since the preference signals are weak and the policy cannot distinguish between the preferred and the dispreferred response. Consequently, Lemma 1a, with the correction, leads to the same conclusions. We have made this correction in the draft, including in the penultimate statement in the proof of Lemma 1b (Appendix A.2). While this expression contained a typographical error that stated that \\( log \left( \frac{\pi_{\theta^*}(y \mid x)\pi_{\mathrm{ref}}(y' \mid x)}{\pi_{\theta^*}(y' \mid x)\pi_{\mathrm{ref}}(y \mid x)} \right) \to 0 \\) (now corrected), Lemma 1b in the main body already correctly stated that  \\( \left( \frac{\pi_{\theta^*}(y \mid x)\pi_{\mathrm{ref}}(y' \mid x)}{\pi_{\theta^*}(y' \mid x)\pi_{\mathrm{ref}}(y \mid x)} \right) \to \infty \\)   from \\( log \left( \frac{\pi_{\theta^*}(y \mid x)\pi_{\mathrm{ref}}(y' \mid x)}{\pi_{\theta^*}(y' \mid x)\pi_{\mathrm{ref}}(y \mid x)} \right) \to \infty \\) .The two expressions are now consistent.
>
> The proof of Lemma 2 in the submitted draft likewise contained an unintentional typographical error stemming from writing conclusions from the \pi log-ratios in Lemma 1 **with the log operator** when they should have been written without it. In this case, when preference probabilities are assigned as 1 for non-deterministic preference samples (which is more likely since there are very few samples to correctly estimate it from), the estimated reward differences tend to positive infinity. This leads to both \\( \log \left( \frac{\pi_{\theta^*}(y \mid x) \pi_{\mathrm{ref}}(y' \mid x)}{\pi_{\theta^*}(y' \mid x) \pi_{\mathrm{ref}}(y \mid x)} \right) \to \infty \\) and \\( \frac{\pi_{\theta^*}(y \mid x) \pi_{\mathrm{ref}}(y' \mid x)}{\pi_{\theta^*}(y' \mid x) \pi_{\mathrm{ref}}(y \mid x)} \to \infty \\) where the second term/equation is what we already stated in Lemma 1b in the main body. So, the conclusion in this case continues to hold after fixing the typo. We believe this should satisfy your concern since after this correction, all expressions should be consistent and the rest of the proof follows the same line of argument starting from Eq. 19 in the paper.
>
> **Clarification of Figure 2:**
>
> On the y-axis we plot the value of the **contrastive log-unlikelihood** (DRDO’s preference component—see last paragraph of Sec. 4) and provide trajectories of this value vs. the log-unlikelihood ratio. This illustrates how $p_w$, which estimates the model’s confidence in its preference assignment, provides an adaptive penalty: if the model is already confident ($p_w = 1$), the loss penalty is less compared to instances where $p_w$ is low. In these cases, gradient updates are amplified by the modulating focal-softened term, enabling the model to still learn with non-deterministic samples (when the model’s estimates are approximately $\frac{1}{2}$).
>
> This is stated in the last paragraph of Sec. 4. We have changed the label on the y-axis of Fig. 2 to read “Contrastive Log-unlikelihood” to make clearer what the plot is showing, and as stated in the text, this is the DRDO preference component of the complete loss, as opposed to the reward difference component..

---

> ### Author Response · Authors · 2024-11-21
> **Pluralistic preferences/DRDO's distinction**
>
> **DRDO vs. pluralistic preferences:**
>
> In certain circumstances, non-deterministic preferences, as reflected in low labeler confidence or equal rewards, could be a consequence of innate pluralistic tendencies of human preferences. However, DRDO is not motivated directly by pluralistic preferences, where there are multiple annotations (or preferences) for a single $(x, y_1, y_2)$, but by the diversity of preference *strength* for paired samples. Typically, pluralistic approaches require multiple reward models (rewarded soups [1], e-DPO [2], MaxMin-RLHF [3], or conditioned policy [4]) to model such preferences. This is computationally expensive and *assumes rewards over multiple dimensions can be linearly interpolated*. We do not make any such assumptions: 1) our main argument is that non-deterministic preferences likely constitute a non-trivial amount of paired samples in popular preference datasets and as such, DRDO provides an efficient alignment method under such conditions. 2) our only strong assumption in the modeling is that the Oracle reward model, given sufficient data, should reasonably approximate human preferences using any standard reward-modeling approach. Furthermore, given such an oracle, we directly regress on the rewards and, unlike e-DPO, do not need to find additional optimal parameters like $\beta$ in the regression. Thus do not need to learn a variety of models each unique to specific viewpoints expressed in the data, and thus our results that best the competitor baselines reflect that we are able to fit better to non-deterministic preferences while still maintaining an ability to fit to deterministic preferences and the data distribution at large.
>
> [1] Rame, A., Couairon, G., Dancette, C., Gaya, J. B., Shukor, M., Soulier, L., & Cord, M. (2024). Rewarded soups: towards pareto-optimal alignment by interpolating weights fine-tuned on diverse rewards. Advances in Neural Information Processing Systems, 36.
>
> [2] Fisch, A., Eisenstein, J., Zayats, V., Agarwal, A., Beirami, A., Nagpal, C., ... & Berant, J. (2024). Robust preference optimization through reward model distillation. arXiv preprint arXiv:2405.19316.
>
> [3] Souradip Chakraborty, Jiahao Qiu, Hui Yuan, Alec Koppel, Dinesh Manocha, Furong Huang, Amrit Bedi, and Mengdi Wang. 2024. MaxMin-RLHF: Alignment with Diverse Human Preferences. In ICML. https://proceedings.mlr.press/v235/chakraborty24b.html
>
> [4] Kaiwen Wang, Rahul Kidambi, Ryan Sullivan, Alekh Agarwal, Christoph Dann, Andrea Michi, Marco Gelmi, Yunxuan Li, Raghav Gupta, Avinava Dubey, et al. Conditioned language policy: A general framework for steerable multi-objective finetuning. arXiv preprint arXiv:2407.15762, 2024.
>
> **Lemma 1 vs. results from existing literature:**
>
> As stated in the lemmas, our argumentation draws substantially from the IPO [1] and e-DPO [2] papers. However, as stated, our Lemma 1 derives from Proposition 1 for cases with zero and near-zero reward differences. This case is never presented in either the e-DPO or IPO papers, and as such our analysis constitutes a novel contribution. We agree that the arguments in Lemmas 1b and 1c are similar in spirit to some arguments in [1] and [2] and we state as such, however the crucial difference is that these Lemmas also follow from Proposition 1 and, with the corrections made to the proof and the demonstration that the conclusions of Lemma 1 still hold given the corrections, these both expose new results for the case of non-deterministic preferences. Lemma 1c, in particular, is not brought to light by either [1] or [2], and Lemma 2 is the first time that similar limitations are exposed in e-DPO. As such, we hope the motivation for DRDO is clear: we provide an efficient, non-ensemble and reference-free method for preference alignment that avoids limitations in two major current methods. We have added this statement to the introduction in the revised paper.
>
> [1] Mohammad Gheshlaghi Azar, Mark Rowland, Bilal Piot, Daniel Guo, Daniele Calandriello, Michal Valko, and Rémi Munos. A general theoretical paradigm to understand learning from human preferences, 2023.
>
> [2] Adam Fisch, Jacob Eisenstein, Vicky Zayats, Alekh Agarwal, Ahmad Beirami, Chirag Nagpal, Pete Shaw, and Jonathan Berant. Robust preference optimization through reward model distillation, 2024. URL https://arxiv.org/abs/2405.19316.

---

> ### Author Response · Authors · 2024-11-21
> **Ablations with IPO/distillation/contrastive component in DRDO**
>
> **Ablations with IPO/distillation/contrastive component in DRDO:**
>
> Multiple reviewers (**Pipq**, **G9Er**, **z2iA**) asked about the quality of the oracle reward model as well as ablations without the reward distillation and the contrastive component in DRDO, we present additional results below where we use the GPT-4o model for win-rate evaluation against these ablated baselines on CNN daily dataset. **Arguably, since GPT-4o is a much stronger/larger model with presumably more data coverage, it should be a reasonable proxy for a closer-to-oracle model**. We randomly sampled 40 prompts from the CNN daily test set (OOD) and compute win-rates and reward margins of samples generated with a top-p of 0.8 and T=0.7 for all baselines vs. DRDO policies trained on Reddit TL;DR using the Phi-3-Mini-4K-Instruct model. We include the IPO baselines [1] (with $\beta$  or $\tau$ = 0.1), the baselines w/o the distillation ( DRDO (-R)) and contrastive component (DRDO (-C)). Additionally, we include the baseline where reward distillation term in DRDO is replaced by the DPO loss but keep the contrastive log-unlikelihood component (DPO (-C)) as asked by **z2iA**. Note that due to compute constraints, we only train these policies from the SFT checkpoint for 500 steps with an effective batch size of 128. For the reward estimates using GPT-4o, we only add one additional condition to the prompt shown in Fig. 5 to get scalar rewards (between 0 and 1). We have added this table and the slight prompt modification in appendix E in our revised draft.
>
>
> | Comparison              | WR A (%) | WR B (%) |  Reward A               | Reward B               |  Margin A           | Margin B           |
> |-------------------------|----------|----------|-----------------|-----------------|-----------------|-----------------|
> | DRDO vs. DRDO (-R)       | 85.0     | 15.0     | 0.24 ± 0.26     | 0.07 ± 0.08     | 0.22 ± 0.22     | 0.09 ± 0.04     |
> | DRDO vs. DRDO (-C)       | 90.0     | 10.0     | 0.25 ± 0.23     | 0.05 ± 0.07     | 0.23 ± 0.22     | 0.06 ± 0.03     |
> | DRDO vs. IPO             | 65.0     | 35.0     | 0.21 ± 0.17     | 0.21 ± 0.24     | 0.09 ± 0.08     | 0.16 ± 0.16     |
> | DRDO vs. DPO (-C)            | 80.0     | 20.0     | 0.18 ± 0.16     | 0.14 ± 0.15     | 0.11 ± 0.08     | 0.22 ± 0.21     |
> Table shows DRDO policies (shown as A) compared against various baselines (shown as B). Win-rates (WR) are computed using average of reward comparison for each sample and then averaged. Margins are computed using the difference of rewards. These ablations suggest that DRDO’s reward distillation as well as the contrastive log-unlikelihood components are crucial for high-quality generations as rated by a stronger oracle like GPT-4o. This is expected since DRDO (-R) has no likelihood-based preference component and is essentially approximating a standard reward model without conditioning it for language generation [2]. This likely leads to its low performance compared to our proposed DRDO loss. OTOH, we observe that solely keeping the log-unlikelihood component does not bring any additional improvement over DRDO. Interestingly, the win-rate over IPO is relatively moderate but substantial over DRDO replaced the DPO term in the first term but with higher margins and variability in both baselines over DRDO.
>
>
>
> [1]  Mohammad Gheshlaghi Azar, Mark Rowland, Bilal Piot, Daniel Guo, Daniele Calandriello, Michal Valko, and Rémi Munos. A general theoretical paradigm to understand learning from human preferences, 2023.
>
> [2] Yang, Rui, et al. "Regularizing Hidden States Enables Learning Generalizable Reward Model for LLMs." arXiv preprint arXiv:2406.10216 (2024).

---

### Official Review · Reviewer_Pipq · 2024-10-30

**Soundness:** 3
**Presentation:** 2
**Contribution:** 2
**Rating:** 5
**Confidence:** 3

**Summary:**

Preference-based fine-tuning is known as one of the most effective methods to boost the generation ability of the modern large-language model (LLM). Direct Preference Optimization (DPO) provides a reasonable solution to this problem. However, the author points out that due to the design of the objective function and the non-deterministic preference label, the trained policy may degenerate, which can be treated as a kind of underfitting. To this, the author suggests DRDO (Direct Reward Distillation and policy-Optimization), which models rewards and preferences simultaneously. They adopt an oracle model that reflects the true preference and distillate its knowledge to a student model. Their method significantly outperforms the common baseline, and their analysis also matches the intuition.

**Strengths:**

Their concern is quite intuitive and handles the fundamental limitation of PbRL. Especially, they mainly point out that the human's preference label may be both noisy and stochastic; hence, the trained model may be confused by the ambiguous update signal. They formulate their intuition into a mathematical lemma and verify its validity. The experimental gap is also significant, and it emphasizes the effectiveness of the proposed method.

**Weaknesses:**

Even though their mathematical formulation and lemmas, their core algorithmic procedure is mainly written in natural language rather than pseudo-code, which diminishes the understandability of the overall process. On the extent of this problem, the components, techniques they adopted, and their main concerns are not harmonic. For example, to the best of my understanding, their key claim is that the 0-1 binary label fails to reflect the uncertainty of the human's preference, and therefore, the trained policy model can be either overfitted or underfitted. Consequently, they suggest to modify the objective function into a regression form rather than the MLE form. However, the concept of the oracle model, knowledge distillation, was suddenly mentioned. My understanding is that they train an Oracle preference model based on the given preference label, distillate its information to the student model, and use the student model for policy optimization. I am not sure how this idea handles the pointed limitation of DPO.

**Questions:**

1. In the common sense of machine learning research, the term 'oracle' usually sounds impossible to achieve, only used during the upper bound experiment or thought experiment. I understand that you 'trained' an Oracle model from the given preference dataset. Is the coverage of the dataset enough to say your trained model is an oracle? Further, isn't your concern mainly built on the low data coverage assumption (Assumption 1)? Please provide more explanation on this part.

2. You said that because the dataset coverage is not infinite but limited, OOD works as a critical drawback to the PbRL framework. How does your framework handle this issue? What feature of knowledge distillation allows us to overcome the low data coverage?

3. I understand that your work mainly tries to handle the fuzzy, unambiguous label setting. Indeed, you split the dataset into a relatively fuzzy subset and a clear subset (hc,he & lc,le). However, the performance gap provided in Table 1 is pretty analogous, regardless of the subset partition. In my intuition, it indicates at least one of these: the edit-distance and human-confidence-based partitioning was not proper, or the performance gap comes from the other intuition rather than fuzzy-label handling. Would you provide more clarification or other experimental results on this point?

**Details Of Ethics Concerns:**

.

---

> ### Author Response · Authors · 2024-11-21
>
> **Pseudo-code for algorithm**
> We thank you for your helpful comments!
> Thank you for pointing this out. While we acknowledge the need for more clarity regarding the DRDO steps/algorithm, we chose natural language to represent it since we assumed the theoretical section required more formalism than the training/initialization steps for DRDO. Note that we also provide full code implementation for DRDO with documentation in our supplementary material. However, we do see the benefit of having a pseudo-code outline of DRDO in the paper. As such, we will include the DRDO algorithm in the appendix.
>
> **Binary labels/Overcoming DPO’s limitations**
>
> We would like to clarify that our main argument is that typical preference datasets contain a non-trivial amount of non-deterministic preference samples apart from those that fall within the spectrum of preference probabilities. However, we point out that a major limitation in DPO is that it models these preference probabilities with a straightforward Bradley-Terry assumption i.e., its sample level loss solely depends on the reward-difference based implicit reward formulation between winning and losing responses without additional regularisation on the policy. This causes it to underfit *not* just for deterministic samples [2, 3] and similar responses[1] but also for non-deterministic samples (a case which is not presented in related prior work unless we are missing something). This is why in our experiment on TL;DR, we focus on both deterministic (high-labeler confidence) as well non-deterministic (low-edit distance in responses) and their combinations to precisely show where DRDO brings an improvement over DPO, as evidenced in our empirical results.
>
>
> Additionally, we prove in Lemma 1 that even the $\beta$ parameter in DPO that controls the policy’s regularisation is not sufficient in such cases. This clarifies our motivation: DRDO’s reward distillation is regression-based (automatically avoids the instability of logs at extremes) along with a contrastive log-likelihood term that provides a sample-level adaptive penalty in the DRDO loss–that makes it learn from  *all* preference samples (on the spectrum). Also, as outlined in our response to **aE9X**, DRDO explicit reward formulation allows it to explore a varied reward space whereas DPO’s reward is sparse [3] (bandit-setting)--this, along with the additional constraint on DRDO to adaptively boost (suppress) winning (losing) response likelihoods leads to a more optimal learning across the spectrum of preference strengths.
>
> **Introducing Oracle/ Knowledge distillation**
>
> We clearly state, in our abstract as well as in the introduction (Fig. 1) and in sec. 4, the crucial role that the oracle plays in DRDO. Furthermore, we also talk about preference learning objectives in related work with a specific focus on a major knowledge distillation based objective (e-DPO) [2], that we exhaustively compare our method to in our empirical results along with theoretical results on [2]’s limitations.
>
> **Oracle sufficiency/ data coverage**
>
> We appreciate your point! To clarify, our assumption 1 clearly states that we assume *sufficient* coverage in the preference dataset. However, it is very challenging to train a true oracle on full-coverage data since in reality getting preference annotations is expensive and time-consuming, even with an LLM. As such, almost all prior academic work focuses on typical preference datasets that contain limited amounts of data-points per example, as also argued in [2, 4]. We take this same approach and go about training the oracle on the full-scale of data available per dataset.For example, we let the oracle train on the entire TL;DR training data ($\mathcal{D}{all}$) even during DRDO alignment for $\mathcal{D}{hc,he}$ and $\mathcal{D}_{\ell c,\ell e}$. Additionally, within our compute budget, we chose larger models (like OPT 1.3B andPhi-3-Mini-4K-Instruct) for oracles compared to policies like OPT 125/350M–while leveraging the expressivity of larger oracles in full-scale (without PEFT) for clear analysis. However, to further resolve this concern we present new results with much more high-capacity/larger (GPT-4o) for reward evaluation for with specific ablation on DRDO’s reward and preference components in our response to **G9Er**. Unfortunately, we could not conduct full-scale training with GPT-4o as an oracle during training due to time/expense reasons. However, these results show that DRDO consistently outperforms baselines when evaluated with a stronger oracle. We hope this should resolve your concern.

---

> > ### Author Response · Authors · 2024-11-21
> > **Additional comments**
> >
> > **How knowledge distillation resolves the OOD problem:**
> >
> > We appreciate your comment on this. We would first like to distinguish between our current work and PbRL frameworks. Standard PbRL-based frameworks [5, 6] typically do not require explicit reward learning since they primarily focus on pairwise comparison signals during training. OTOH, DRDO uses explicit reward signals for single options (LLM responses) while still leveraging preference-based signals in the contrastive log-unlikelihood term. As such, our work acts as a bridge connecting PbRL frameworks to typical RLHF-based settings.
> >
> > Note that we use the phrase “OOD” in two sections in the paper–in the oracle training and our DRDO policy evaluation sections only–for *two distinct* reasons. We do not make any claims that our alignment algorithm (eq. 6) in DRDO is motivated to resolve the different types of OOD issues typical in RLHF (distribution shifts in prompts, data, etc). In fact, our empirical experiments reveal that DRDO tends to work better with a new data distribution (CNN Daily evaluation as well as AlpacaEval) than baselines and the reason for including this CNN Daily experiment is to directly compare with [8].
> >
> > The second reason for using the OOD phrase is during oracle training: we want the oracle to generalize well to OOD (data distribution) settings and prevent reward hacking, while being compute-efficient. The usual choice here is to use reward ensembles or their linear parameter interpolation. However, within our compute budget, the most veritable choice is [7]’s method of reward model training that requires *no* ensembles and is architecturally efficient. This uses as additional SFT-based regularisation (eq. 5) that is empirically proven to generalize well without distorting the base LM features [6] and offer consistent rewards. Since our method is entirely *offline*, such an oracle offers high-fidelity rewards which is crucial for careful exploration of the reward space sans the conservativeness offered by ensembles. This also motivates our comparison with e-DPO which is offline but uses reward ensembles for a conservative estimate.
> >
> > **Performance on deterministic vs non-deterministic samples**
> > Thank you for your insightful comment. Firstly, we would like to clarify that our main motivation for DRDO is to design an alignment algorithm that efficiently learns the *full* spectrum of preferences, including deterministic and nondeterministic preference pairs. As outlined in lemma1 and 2 and sec. 4, DRDO avoids policy degeneracy and underfitting issues for both deterministic as well as nondeterministic samples–which is precisely what our experiments on TL;DR summarization show.
> > **Ambiguous labels/ analogous performance on TL;DR**
> > The motivation for including both labeler confidence and string-similarity of pairs to simulate this setting is also clarified in our response to **z2iA** performance. In particular, there’s a subtle but crucial distinction in how the literature treats noise in preference annotations–prior work [9, 10] assumes this noise is present in flipped labels and requires either heuristics, data exclusion or prior knowledge of noise coefficients in the data for reward modeling. We do not make such assumptions and DRDO does *not* require any data exclusion during training and our $\gamma$ parameter in eq. 6 is in practice a cross-validated hyperparameter though provide more insights in our response to **aDcH**. In fact, we argue that using labeler confidence and string-similarity of pairs is intuitive since both can be a function of how clear the preference signal is. As such, although we agree that the win-rate performance across these difficulty-based subsets are close, they are consistently close to 80 percent.  This suggests that DRDO is robust to all settings without sacrificing preference learning in general settings (full data) and is able to learn all types of preference samples–which is precisely how we motivate DRDO.

---

> > > ### Author Response · Authors · 2024-11-21
> > > **Citations for above comments**
> > >
> > > We present the cited work below:
> > >
> > >  [1] Pal, A., Karkhanis, D., Dooley, S., Roberts, M., Naidu, S., & White, C. (2024). Smaug: Fixing failure modes of preference optimisation with dpo-positive. arXiv preprint arXiv:2402.13228.
> > >
> > > [2] Adam Fisch, Jacob Eisenstein, Vicky Zayats, Alekh Agarwal, Ahmad Beirami, Chirag Nagpal, Pete Shaw, and Jonathan Berant. Robust preference optimization through reward model distillation, 2024. URL https://arxiv.org/abs/2405.19316.
> > >
> > > [3] Rafailov, R., Hejna, J., Park, R., & Finn, C. (2024). From $ r $ to $ Q^* $: Your Language Model is Secretly a Q-Function. arXiv preprint arXiv:2404.12358.
> > >
> > > [4] Mohammad Gheshlaghi Azar, Mark Rowland, Bilal Piot, Daniel Guo, Daniele Calandriello, Michal Valko, and Rémi Munos. A general theoretical paradigm to understand learning from human preferences, 2023.
> > >
> > > [5] Wirth, C., Akrour, R., Neumann, G., & Fürnkranz, J. (2017). A survey of preference-based reinforcement learning methods. Journal of Machine Learning Research, 18(136), 1-46.
> > >
> > > [6] Munos, R., Valko, M., Calandriello, D., Azar, M. G., Rowland, M., Guo, Z. D., ... & Piot, B. Nash Learning from Human Feedback. In Forty-first International Conference on Machine Learning.
> > >
> > > [6] Ananya Kumar, Aditi Raghunathan, Robbie Jones, Tengyu Ma, and Percy Liang. Finetuning can distort pretrained features and underperform out-of-distribution. arXiv preprint arXiv:2202.10054, 2022.
> > >
> > > [7] Yang, R., Ding, R., Lin, Y., Zhang, H., & Zhang, T. (2024). Regularizing Hidden States Enables Learning Generalizable Reward Model for LLMs. arXiv preprint arXiv:2406.10216.
> > >
> > > [8] Rafailov, R., Sharma, A., Mitchell, E., Manning, C. D., Ermon, S., & Finn, C. (2024). Direct preference optimization: Your language model is secretly a reward model. Advances in Neural Information Processing Systems, 36.
> > >
> > > [9] Chowdhury, S. R., Kini, A., & Natarajan, N. (2024). Provably robust dpo: Aligning language models with noisy feedback. arXiv preprint arXiv:2403.00409.
> > >
> > > [10] Wang, B., Zheng, R., Chen, L., Liu, Y., Dou, S., Huang, C., Shen, W., Jin, S., Zhou, E., Shi, C., et al. Secrets of rlhf in large language models part ii: Reward modeling. arXiv preprint arXiv:2401.06080, 2024.

---

> ### Comment · Reviewer_Pipq · 2024-11-25
> **Comment**
>
> Thank you for your detailed clarification. I think the authors' rebuttal covers a slightly different area than my concerns. I will maintain my original score.

---

> > ### Author Response · Authors · 2024-11-25
> >
> > Thank you for your reply. Is it possible to expand on where our response diverges from the issues you originally raised?

---

> > > ### Author Response · Authors · 2024-11-26
> > >
> > > Dear reviewer,
> > >
> > > As the discussion period is coming to an end, we'd like to address your concerns in light of new experimental results. In particular wrt to your comment on analogous results on TL;DR training partitions, our observation is that DRDO is robust to all difficulty settings including partitions containing significant non-deterministic samples and it operates without sacrificing preference learning in general settings (full data). However, we provide additional results with increasing OOD extent (with input prompt length as proxy for OOD) to see the distribution of win-rates as OOD increase. The results here (Appendix E.2 and Fig. 5 in our revised paper) clearly show that DRDO wins over e-DPO and DPO, on average, tend to show an increasing trend as OOD quantification increases.
> > >
> > > We believe this result should resolve your question of how explicit reward distillation (knowledge distillation) resolves the low-data coverage in PbRL in OOD settings--> implicit rewards (either DPO or regression in e-DPO) are essentially modeled as a conditional distribution over prompts ($x$) which is 1) sparse due to its bandit setting and 2) where their dependence on prompts is effectively cancelled out due to the Bradley-Terry assumption (log-partition term cancellation in deriving both DPO and e-DPO papers). OTOH, given a strong enough oracle, explicit reward computation is over the joint-distribution of both prompts ($x$) and responses ($y$) (as seen in eq. 5 and 6 in our paper)--> this naturally captures a richer interaction between prompts and preferred (or dispreferred) responses, given the expressivity of LLMs chosen for our experiments with billions of parameters. As such, when the input prompt distribution changes (our OOD experiment), DRDO's explicit reward modeling (after having explored a richer reward space and interactions between prompt and response variables during training) is able to  generalize well and generate more high-quality summaries precisely due to it learning these interactions.
> > >
> > > As as result, DRDO policies end up learning specific correlations between structures present in reddit posts (as prompts) and how it affects the preference in the summaries; while DPO and e-DPO are only constrained to maximize its reward gap given the prompt where the interactions with the prompt is effectively cancelled by definition. Furthermore, upon a closer manual inspection, we find that in this case DRDO summaries are on average more succinct and relevant, without adding repetitive details typical in news articles contain, that DPO as well as e-DPO summaries tend to do. This can explain DRDO’s high performance in this OOD setting and throw light on how it resolves the low-data coverage in PbRL in OOD settings.
> > >
> > > Lastly, we have also included a summary of our full set of revisions and additional experiments/results with a comment at the top. Please let us know if you have any additional concerns and we'd be happy to resolve them! Once again, thank you for your time and effort in reading and reviewing our paper!

---

### Official Review · Reviewer_aDcH · 2024-11-01

**Soundness:** 3
**Presentation:** 3
**Contribution:** 3
**Rating:** 5
**Confidence:** 4

**Summary:**

The paper introduces Direct Reward Distillation and policy-Optimization (DRDO), a novel approach to preference alignment that combines reward modeling and policy optimization into a unified framework. Traditional methods like Direct Preference Optimization (DPO) can struggle with preference ambiguity, leading to suboptimal policies. DRDO addresses these issues by directly distilling rewards from an Oracle into the policy model while simultaneously learning from diverse preference signals. DRDO is evaluated on multiple benchmarks (Ultrafeedback, TL;DR, and AlpacaEval) and shows improved robustness, especially in handling noisy or non-deterministic preferences, outperforming popular methods like DPO and e-DPO.

**Strengths:**

* Sound Theoretical Foundation: The paper provides a clear theoretical grounding, with detailed proofs supporting DRDO’s robustness to varied preference strengths.

* Experimental Quality: The experiments are well-designed, covering both deterministic and non-deterministic preference data, and include benchmarks that highlight DRDO’s superiority across multiple settings and model sizes.

**Weaknesses:**

* A bit concern on novelty and motivation: this paper seems to combine the ideas from three papers: eDPO, Oracle RM(https://arxiv.org/abs/2406.10216), and Focal. The main motivation and contribution need to be better justified in order to be a high quality paper in ICLR.

* Out-of-distribution (OOD) concern: For OOD experiment, it is a bit unclear to me on what the authors want to address. There are a few distribution shifts involved in alignment: 1. prompt shift among reward model training data, rlhf training prompt, and rlhf evaluation prompt. 2. response shift between the reward model training data and on-policy generation. As far as I understand, the TL;DR dataset also contains a few CNN Dailymail examples. It will be helpful if authors can show how robust the proposed approach can address the OOD issue systematically. Say show a plot with x-axis (the amount of OOD) and y-axis (the performance gap between baseline and proposed approach).

* Additional complexity of the system: focal loss introduces additional two hyperparameters, it is challenging to tune them, especially for OOD. What evaluation dataset to use? How to make sure the evaluation dataset is not overfitted and represent a good coverage of OOD samples during testing?

**Questions:**

See weakness.

---

> ### Author Response · Authors · 2024-11-21
> **Robustness to OOD settings/additional experiments**
>
> **Robustness to OOD settings:**
> Thank you for your insightful comments.
> Apologies for the confusion–we’d like to start off by clarifying that for the summarization task, we do not train DRDO or any baselines on the CNN daily articles (eval set) and only train on the Reddit TL;DR posts, the former being an entirely unseen data distribution. This is what we meant by testing on OOD settings, as in with a new input (prompt) distribution. This is to directly compare with baselines like DPO as this is a popular distribution to test this aspect [1]. We already outlined this difference in section 5 (second paragraph) but we understand this might have caused confusion–as such, we will add a line to clarify this in our revised version.
>
> Within this definition of OOD, as requested, we conduct a more systematic experiment to test OOD generalisation ability of DRDO vs baselines using article- lengths as proxy for OOD-composition. To test this,we randomly sample 1,000 prompts from this dataset and segment these prompts into bins of 50 tokens based on prompt-token counts, spanning the full range of token lengths. We then sample completions from all baselines including DRDO using a top-p of 0.8 and temperature of 0.7. For this automatic evaluation of summary quality, we use GPT-4o as a high-capacity judge using the same prompt as shown in Fig. 6. This evaluation effectively measures the OOD generalization capabilities of the policies as prompt lengths (and corresponding news article lengths) increase. For this automatic evaluation, we use GPT-4o as a high-capacity judge The results are given in the table below:
>
> The trends in the table below suggest that as OOD composition (with prompt-token lengths as proxy) increases, on average DRDO policies tend to have relatively larger win-rates compared to shorter prompts over baselines like DPO and e-DPO. In contrast, we find that DPO and e-DPO win-rates tend to decrease with increase in prompt-lengths. We have added this result as a plot in appendix E in our revised paper.
>
>
> | Avg Prompt Length | DRDO vs DPO (WR %) | DPO vs DRDO (WR %) | DRDO vs e-DPO (WR %) | e-DPO vs DRDO (WR %) |
> |-------------------|-----------------------|-----------------------|------------------------|------------------------|
> | 193.80            | 90.00                | 10.00                | 60.00                 | 40.00                 |
> | 247.58            | 72.22                | 27.78                | 78.95                 | 21.05                 |
> | 302.88            | 70.97                | 29.03                | 71.88                 | 28.12                 |
> | 355.72            | 78.57                | 21.43                | 78.57                 | 21.43                 |
> | 404.53            | 80.85                | 19.15                | 78.00                 | 22.00                 |
> | 451.95            | 87.18                | 12.82                | 75.64                 | 24.36                 |
> | 501.51            | 87.84                | 12.16                | 81.08                 | 18.92                 |
> | 558.28            | 83.05                | 16.95                | 88.71                 | 11.29                 |
> | 609.92            | 79.12                | 20.88                | 76.34                 | 23.66                 |
> | 653.77            | 78.63                | 21.37                | 78.51                 | 21.49                 |
> | 705.55            | 77.78                | 22.22                | 82.18                 | 17.82                 |
> | 755.17            | 84.42                | 15.58                | 79.01                 | 20.99                 |
> | 801.56            | 72.62                | 27.38                | 77.22                 | 22.78                 |
> | 842.69            | 85.19                | 14.81                | 81.82                 | 18.18                 |
> | 889.17            | 91.67                | 8.33                 | 90.00                 | 10.00                 |

---

> ### Author Response · Authors · 2024-11-21
> **novelty and motivation**
>
> **Novelty and motivation**
> Our novelty lies in specifically identifying underfitting and policy degeneracy in two popular preference learning algorithms--DPO with its focus on implicit rewards and e-DPO with explicit rewards--and providing new theoretical insights on these limitations for both deterministic and non-deterministic preferences. Crucially, the training suboptimality and consequent policy quality of DPO (a very popular algorithm at this point) for deterministic preference cases is already well-studied in the literature [1-4]. However, very few works have focussed on the non-deterministic scenario–even though such samples are likely non-trivially present in popular preference dataset. *In offline training with limited data, it is therefore crucial to develop approaches that are robust to both types of preferences.*
>
> As such, we do point out e-DPO’s limitations in lemma 2 and unlike e-DPO, DRDO does not constrain its explicit reward learning with the KL-optimal policy based implicit rewards [5]. This is consistent with DRDO’s choice of an unlikelihood-based regularization that does not require a separate reference policy but learns adaptively from the spectrum of preferences, including the extremal points. This adaptive penalty is our primary motivation for using a focal modulating term but our formulation is entirely new and is unlike how the focal term is used traditionally [7]. As outlined in sec. 4,  $p_w$ is the model’s estimate of p* (true preference probabilities) and do not imply class probabilities as is typical in the focal loss literature.
>
> Our novelty lies in adapting this sample-wise modulating term for preference alignment with a one-step sigmoid computation over the log-likelihoods. While it is true that this does introduce two additional parameters, our experience as well as our empirical studies suggest that $\alpha = 0.1$ is generally more stable for training across benchmarks. As for the exponential $\gamma$, $\gamma$ = 2 appears to be a reasonable choice, similar to [7]. A larger  $\gamma$ can harshly penalize the loss when the policy is uncertain ($p_w <<1$ ) while a smaller $\gamma$ = 0 may not adequately penalize and impact its adaptive nature. This is evidenced in our additional results in response to **z2iA** where the optimal reward is achieved for $\gamma$ = 2 while too low or too high a $\gamma$ can affect performance. As such, although we find $\gamma$ = 2 to be optimal across datasets, a reasonable way to find the right $\gamma$ would vary case by case–if your has baseline policy at the start of training has undergone no SFT or off-policy settings, a lower $\gamma$ could be ideal since a higher $\gamma$ might apply harsher penalties in this case. However, if starting with SFT (as we do) or on-policy (where $p_w$ is likely to be higher already), a higher $\gamma$ can be optimal.  In practice though, empirical validation on a held-out set can be an efficient alternative, similar to how an optimal $\beta$ can be determined in algorithms like DPO.
>
> **On OOD evaluation dataset**
>
> We would like to clarify that our main motivation is not to retrieve an OOD-generalizable policy since this can depend crucially on the base model (size, pretraining data coverage etc). In fact, generalizability to OOD-settings (new input-prompt distribution) is only one component of our experiments as clarified above. However, we believe popular benchmarks like AlpacaEval for instruction-following or CNN daily for summarization can provide reasonable tests including data coverage for such evaluation. As for overfitting, typical methods like data separation, avoiding leakage etc should suffice to resolve this issue.

---

> > ### Author Response · Authors · 2024-11-21
> > **Additional citations**
> >
> > Here we outline the citations for our previous comment.
> > [1]  Adam Fisch, Jacob Eisenstein, Vicky Zayats, Alekh Agarwal, Ahmad Beirami, Chirag Nagpal, Pete Shaw, and Jonathan Berant. Robust preference optimization through reward model distillation, 2024. URL https://arxiv.org/abs/2405.19316.
> >
> > [2]  Mohammad Gheshlaghi Azar, Mark Rowland, Bilal Piot, Daniel Guo, Daniele Calandriello, Michal Valko, and Rémi Munos. A general theoretical paradigm to understand learning from human preferences, 2023
> >
> > [3] Rafailov, R., Hejna, J., Park, R., & Finn, C. (2024). From $ r $ to $ Q^* $: Your Language Model is Secretly a Q-Function. arXiv preprint arXiv:2404.12358.
> >
> > [4] Pal, A., Karkhanis, D., Dooley, S., Roberts, M., Naidu, S., & White, C. (2024). Smaug: Fixing failure modes of preference optimisation with dpo-positive. arXiv preprint arXiv:2402.13228.
> >
> > [5]  Rafailov, R., Sharma, A., Mitchell, E., Manning, C. D., Ermon, S., & Finn, C. (2024). Direct preference optimization: Your language model is secretly a reward model. Advances in Neural Information Processing Systems, 36.
> >
> > [6] Yang, Rui, et al. "Regularizing Hidden States Enables Learning Generalizable Reward Model for LLMs." arXiv preprint arXiv:2406.10216 (2024).
> >
> > [7] Ross, T. Y., & Dollár, G. K. H. P. (2017, July). Focal loss for dense object detection. In proceedings of the IEEE conference on computer vision and pattern recognition (pp. 2980-2988).

---

### Official Review · Reviewer_aE9X · 2024-11-04

**Soundness:** 2
**Presentation:** 2
**Contribution:** 2
**Rating:** 5
**Confidence:** 4

**Summary:**

This paper introduces Direct Reward Distillation and Policy Optimization (DRDO). The method combines explicit reward modeling with policy optimization by training an oracle model to provide reward signals, then the signals are directly distilled into the policy model. Additionally, DRDO employs a loss function that integrates a reward difference component and a contrastive log-unlikelihood term, adjusting the learning process based on the model’s confidence in assigning preferences.

**Strengths:**

1. DRDO avoids dependence on a reference model, which could offer computational benefits.

2. In experiments, the inclusion of out-of-distribution settings (e.g., using CNN Daily articles) aligns with the claims of improved robustness.

**Weaknesses:**

1. The paper lacks a clear definition of key variables such as $\hat{r}_1$ and $\hat{r}_2$, which are crucial for understanding the proposed loss function. This lack of detail impedes understanding of how the reward signals are computed and integrated into the training process.

2. The authors seem to overstate their contributions; the distillation loss they employ appears to be derived from prior work, specifically the paper “Robust Preference Optimization through Reward Model Distillation” by Fisch et al. (2024). Proper attribution and a clearer differentiation from existing methods are needed.

3. Key concepts, such as the handling of ambiguous preferences and the role of the modulating focal loss component, require more elaboration. Terminologies like “reward difference” and “contrastive log-unlikelihood” could be better clarified to aid understanding.

**Questions:**

1. How are $\hat{r}_1$ and $\hat{r}_2$ defined within the loss function? Providing clear definitions would enhance understanding of the method’s implementation.

2. Given that the distillation loss resembles that in Fisch et al. (2024), how does DRDO differentiate itself from this existing method?

3. Why were ablation studies on the focal loss component or comparisons across a broader range of datasets or evaluation metrics not conducted?

4. Considering the emphasis on practical efficiency, how does the computational overhead of DRDO compare to DPO or e-DPO?

---

> ### Author Response · Authors · 2024-11-21
>
> Thanks for your insightful comments!
>
> **How are the policy-estimated rewards defined?**
> In Section 4, we state that the $\hat r$ terms are the estimated rewards computed from the policy during training—we match pairwise reward differences from this policy to those from the Oracle model. Essentially, we are matching the preference strengths in the policy with the oracle. However, we can see how these two types of rewards can cause confusion. We will add a sentence stating clearly the difference between policy and oracle rewards and clarify that they are different and that they are architecturally very similar.
>
>
> **How is DRDO different from e-DPO?**
> DRDO explores a more varied reward space than e-DPO since the policy’s rewards in DRDO distillation are scalars ($\in \mathbb{R}$) while e-DPO enforces a much stronger constraint that the policy’s *implicit* rewards must align with *explicit* rewards. This varied exploration explains DRDO consistently outperforming e-DPO on multiple benchmarks and with multiple model families across temperatures in our empirical results. e-DPO [1] only reports results for summarization task only and with a single model family.  As our experiments suggest, DRDO is more efficient and generalizable to various tasks and preferences compared to both DPO or e-DPO. As stated in our response to reviewer **G9Er**, we not only fit better to non-deterministic preferences, we do it without sacrificing the ability to fit to deterministic preferences and the overall data distribution.
>
> Also, although both e-DPO and DRDO are based on the idea of explicit reward distillation, *the loss functions and landscapes are substantially different*.  DRDO loss consists of an additional term that applies a novel unlikelihood-based objective with a focal-loss term. To the best of our knowledge, we are the first to propose this specific form of preference regularisation.
>
>
>
> **Computational efficiency: DPO or e-DPO**
> e-DPO requires training reward ensembles to form a confidence set for training the policy. In our experiments, we use 3 reward models to construct this set which makes it roughly thrice as expensive as DRDO training. Like DPO, e-DPO requires a separate reference model to be kept in memory, further increasing compute requirements. DRDO, on the other hand, only requires a trained oracle for distillation. The expected rewards (from oracle) can be precomputed once and a separate reference model does not need to be kept in memory (as shown in Eq. 6). During training, DRDO does require one additional linear head on top of the base LM to predict the reward estimates. This adds a negligible 0.003% more trainable parameters (relative to the language modeling head of base LM Phi-3-Mini-4K-Instruct). During inference, DRDO trained policies do not require this head. *We have added this information  in Appendix F.*
> Additionally, e-DPO directly matches the implicit reward differences to those assigned by the oracle, thus putting too much trust in the oracle. While they do acknowledge this in [1] and motivate the use of reward ensembles for conservative exploration, it is not always easy to design ensembles (the optimal choice of reward combinations, etc.) and this can be prohibitively expensive. Our work here should be seen in the light of light-weight methods [2–5] that require no ensembles, are reference-model free, yet provide efficient regularization during offline alignment with a careful choice of the log-unlikelihood component with the dynamic modulating term. We provide results with various gamma values in the focal-unlikelihood term in the table below. We would also like to clarify that we indeed provide empirical result on three different benchmarks including an OOD setting along with multiple model sizes and families—across all benchmarks, we find that DRDO consistently outperforms e-DPO as well as DPO.
>
> [1] Adam Fisch, Jacob Eisenstein, Vicky Zayats, Alekh Agarwal, Ahmad Beirami, Chirag Nagpal, Pete Shaw, and Jonathan Berant. Robust preference optimization through reward model distillation, 2024. URL https://arxiv.org/abs/2405.19316.
>
> [2] Rafailov, R., Sharma, A., Mitchell, E., Manning, C. D., Ermon, S., & Finn, C. (2024). Direct preference optimization: Your language model is secretly a reward model. Advances in Neural Information Processing Systems, 36.
>
> [3] Yang, Rui, et al. "Regularizing Hidden States Enables Learning Generalizable Reward Model for LLMs." arXiv preprint arXiv:2406.10216 (2024).
>
> [4] Huang, A., Zhan, W., Xie, T., Lee, J. D., Sun, W., Krishnamurthy, A., & Foster, D. J. (2024). Correcting the mythos of kl-regularization: Direct alignment without overparameterization via chi-squared preference optimization. arXiv preprint arXiv:2407.13399.
>
> [5] Shicong Cen, Jincheng Mei, Katayoon Goshvadi, Hanjun Dai, Tong Yang, Sherry Yang, Dale Schuurmans, Yuejie Chi, and Bo Dai. Value-incentivized preference optimization: A unified approach to online and offline RLHF. arXiv:2405.19320, 2024.

---

> > ### Comment · Reviewer_aE9X · 2024-11-25
> >
> > > How are the policy-estimated rewards defined? In Section 4, we state that the
> >  terms are the estimated rewards computed from the policy during training—we match pairwise reward differences from this policy to those from the Oracle model. Essentially, we are matching the preference strengths in the policy with the oracle. However, we can see how these two types of rewards can cause confusion. We will add a sentence stating clearly the difference between policy and oracle rewards and clarify that they are different and that they are architecturally very similar.
> >
> > Can you write down the explicit formula in the revised paper? How to calculate $\hat{r}$?

---

> > > ### Author Response · Authors · 2024-11-25
> > >
> > > Thank you for your response. The calculation of the $\hat r$ values is given in the DRDO pseudocode in Appendix B (Table 5, specifically step (a).ii). This is the forward pass on the policy (parameterized with $\theta$ and $\theta ‘$), with a linear MLP on top of the language model. We will add this to Sec. 4 (after Eq. 6) in a revision to be made tomorrow.

---

> > > > ### Author Response · Authors · 2024-11-26
> > > >
> > > > Dear reviewer,
> > > >
> > > > As the discussion period is coming to an end, we hope our revisions on the paper (sec. 4 and tab. 5) clarified your concerns on the policy reward estimation part. We have included a summary of our full set of revisions and additional experiments/results with a comment at the top. Please let us know if you have any additional concerns and we'd be happy to resolve them! Once again, thank you for your time and effort in reading and reviewing our paper!

---

> > ### Comment · Reviewer_aE9X · 2024-11-27
> >
> > > Also, although both e-DPO and DRDO are based on the idea of explicit reward distillation, the loss functions and landscapes are substantially different. DRDO loss consists of an additional term that applies a novel unlikelihood-based objective with a focal-loss term. To the best of our knowledge, we are the first to propose this specific form of preference regularisation.
> >
> > Q: Can you elaborate more on this? Seems that KTO (https://arxiv.org/abs/2402.01306) also have similar terms.

---

> > > ### Author Response · Authors · 2024-11-27
> > >
> > > Thank you for bringing this up! We’d like to clarify that there two substantial distinctions b/w DRDO and e-DPO that directly affects the loss landscapes
> > > - From a theoretical angle,  DRDO explores a more varied reward space than e-DPO since the policy’s rewards in DRDO distillation are scalars ($\in \mathbb{R}$) and are modeled over the joint-distribution of both prompts ($x$) and responses ($y$) (as seen in eq. 6 and table. 5 in our paper). This is precisely what the policy learns to match with the oracle. OTOH, e-DPO enforces a much stronger constraint that the policy’s *implicit* rewards must align with *explicit* rewards, where the implicit rewards are conditionally modeled (where the dependence on the prompt reflected in the log-partition term is canceled out, similar to DPO derivation). **This richer interactions between prompts and responses that DRDO captures with its jointly modeled rewards naturally changes the loss landscapes in these two methods**. In fact these interactions that DRDO affords arguably lead to it consistently outperforming e-DPO across the two tasks, as well as benchmarks  like AlpacaEval—while fixing the size of the oracle.
> > > - Similarly, the novel contrastive log-unlikelihood term with a focal modulating component (second term in eq. 6 in our paper) is different from e-DPO’s pessimistic forward KL-divergence of the reference model over the policy. This former term’s regularization affect in DRDO is adaptive, requires no reference-model and ensures that learning continues in DRDO at all preference strengths (we explain this in depth in sec. 4 as well as to our response to **z2iA**). As such, the motivations for regularization in e-DPO and DRDO are clearly different: in e-DPO, this is to account for uncertainty in reward ensembles with pessimism in learning whereas in DRDO, it is to adaptively learn from the entire spectrum of preferences.
> > > - **Difference with KTO**: we do not find any similarities in DRDO with KTO other than that both are intended for preference alignment. In fact, they are very fundamentally different. KTO is motivated to learn from unpaired binary data and assumes that all samples can be binned into desirable and undesirable samples. OTOH, DRDO makes no such assumptions but considers preferences form a spectrum. Additionally, KTO’s loss requires implicit reward computations (eq. 8 in KTO paper) apart from KL-divergence approximations of the baseline reference point ($Z_o$)—both of which requires a reference model while DRDO does not.

---

> > > > ### Comment · Reviewer_aE9X · 2024-11-30
> > > >
> > > > Thanks for your detailed response, I have raised my score from 3 to 5.

---

> > > > > ### Author Response · Authors · 2024-11-30
> > > > > **Thank you for raising the score**
> > > > >
> > > > > Dear reviewer,
> > > > >
> > > > > We'd like to thank you taking the time in considering our detailed clarifications and for raising your scores. As the discussion period is almost coming to an end, we'd sincerely hope to have further engagement with all reviewers during the rest of the discussion period.
> > > > >
> > > > > Thanks,
> > > > > Authors

---

### Official Review · Reviewer_z2iA · 2024-11-05

**Soundness:** 3
**Presentation:** 4
**Contribution:** 2
**Rating:** 6
**Confidence:** 4

**Summary:**

The paper proposes an extension of Fisch et al 2024 work on Reward Distillation involving a regularization objective and demonstrate marginal improvements in terms of empirical benchmarks. There contributions are primarily two fold:

1) When optimizing the reward model (Oracle in their paper) they add an additional regularization objective that ensures the Oracle still assigns high likelihood to the winning generation.

2) The authors include a gradient unlikelihood objective, (which is further explained mechanistically in the appendix). This objective enhances gradients in the direction of the winning response when the winning response is small (and vice versa for the loosing response).

**Strengths:**

1) The paper is on a very relevant area and proposes a simple extension to existing approaches for preference optimization.

2) The empirical experiments are extensive and instill faith in the reviewer of the proposed algorithms merits.

3) The explanation of the unlikelihood objective in Eq 6 in terms of the gradient updates helps clarify better what the proposed algorithm achieves mechanistically.

**Weaknesses:**

A) Notation is a bit hard to follow in the paper, for example L_O represents the Oracle loss, but is there shared parameterization between the Oracle and the Policy? In either case, this should be explicitly notated in the paper. Similarly the variable alpha is shared between the equation 5 and 6, are these the same or different? It is important to be explicit about these differences.

B) The major contribution in the paper over Fisch et al comes from the unlikelihood term in Equation 6. However this is explained only in passing in the main text, if indeed the authors want the focus to be on the importance of this term.

C) My most major concern of the paper is that it is hard to decouple if the improvements come from i) Improved reward Distillation ii) The unlikelihood objective in Eqn 6) . I would be curious to see if replacing the first term in objective in Eq 6 with a standard DPO objective still results in an improved policy. If so, this framework would be much more generalizable.

**Questions:**

A) In Eq. 5 is the \log(y_w) term \log(y_w|x) or \log(y_w, x) ? This could  have potential ramifications for the quality of the resulting Oracle. In situations where both y_w and y_l are reasonably close generations, would you not want to also penalize the Oracle to be close to \log(y_l|x) ?

B) Was the oracle initialized from an model trained by SFT on the winning responses? This is an important question to understand the oracle regularization and its effects.

C) The proposed objective involves multiple additional hyperparameters, including \alpha in 5, \alpha in 6 as well as \gamma in 6. Can the author's comment on the sensitivity of the model performance to these additional parameters?

---

> ### Author Response · Authors · 2024-11-21
>
> Thank you for your insightful feedback.
>
> **Is there shared parametrization b/w policy and alpha hyperparameter b/w oracle and policy:**
>
> No, there is no shared parametrization between the policy and the Oracle. The $L_{O}$ loss is only a function of the oracle’s own parameters $\phi$ and the preference dataset, $D\_{pref}$. As shown in Eq. 6, our policy training phase is separate with *no* shared parametrization with the Oracle since only policy’s reward estimates are regressed on those from the trained Oracle (Eq. 6 shows that only policy parameters $\theta$ are trained directly on preference data $D\_{pref}$). It is also states in Section 4 in that the Oracle parameters are **not** updated during policy training (and therefore there is no shared parametrization). This is now explicitly stated in Section 4.
>
> Additionally, we clarify the actual value of alpha for Oracle training (Eq. 5) in Section 5 (footnote 5). As such, the $\alpha$ in Eq. 5 and Eq. 6 are different: the former regulates the strength of the SFT regularisation in Oracle training while the latter controls the modulating term in the focal unlikelihood component of policy alignment (Eq. 6). We chose to limit adding new variables terms for clarity and consistency with previous work in reward model training and focal loss literature. We will clarify this with one sentence in section 5. We will also include the exact parametrization of the policy estimates of the reward ($\hat{r}$) in eq. 6 for more clarity.
>
>
> **Importance of the focal/unlikelihood term:**
>
> The log-unlikelihood term provides an adaptive, sample-level penalty in the contrastive component in our $\mathcal{L}_{kd}$ loss that allows for dynamic penalties depending on how confident and correct the policy is for a given sample. The core insight here as explained at the end of sec. 4 is that: unlike DPO’s $\beta$ parameter which is fixed during training, DRDO’s ($(1 - p_w)^\gamma$) term in Eq. 6 applies penalties more intuitively–when the model is already correct and confident ($p_w =1$), the penalty on the loss is minimal but if its substantially wrong ($p_w =0$), the exponential with $\gamma$ applies a hard penalty. OTOH, if p* ~ 1/2 for non-deterministic samples, this term still actively penalises the DRDO loss if the policies estimate of p* with $p_w$ is ~ ½ i.e., a correct estimate. In reality, $p_w$ may *not* be close to ½ for *every* non-deterministic sample in the data but in that case too the gradients in DRDO do what we require which is to increase (decrease) likelihood of the winning (losing) response. In contrast, lemma 1a suggests that DPO’s learning would underfit due to zero implicit reward difference (Bradley Terry assumption). Intuitively, we can also see that the log-unlikelihood term is minimised at convergence where the likelihood of the winning and losing responses are 1 and 0 respectively–which is typically the goal in contrastive preference alignment. Please see our response  to  **aE9X**  for the distinction between e-DPO and DRDO.
>
> **Whether Oracle is initialized with SFT:**
>
> We clearly state in section 4 that our Oracle training is a one-step process: we do not apply a separate SFT training phase for our Oracle reward models as is typical in the literature. However, we do include an SFT term (second term in eq. 5) over the winning responses to regularize the Oracle training for out-of-distribution (OOD) generalization and to avoid reward hacking. The motivations for this are clearly specified in section 4. Note that, we do initialize all our policies (including baselines like DPO and e-DPO) from the SFT-trained model as suggested in related previous work [2, 3] as mentioned in our experimental setup.
>
> **On penalizing the Oracle to be close to $\log(y_l \mid x)$**
> The oracle is simply a reward model that uses Bradley-Terry (BT)-based formulation to learn the rewards. As such, the second term (SFT) in eq. 5 simply regularizes the reward loss [5] since the reward head parameters are randomly initialised (unlike the language generation head which is trained). Since we do not use the oracle for any form of language generation (self-critique etc), the oracle already learns to penalize the losing responses in the BT formulation (first term) in distinguishing rewards between the two responses. Therefore, we do not additionally require the oracle to be close to $\log(y_l \mid x)$
> .
>
> **Improvement over reward distillation vs contrastive log-unlikelihood**
>
> Please check our our response to  **G9Er** with additional ablation results after excluding the distillation and log-unlikelihood term. These results suggest that both these two components are important in our DRDO loss formulation. Explicit rewards help explore a more varied reward space while the contrastive component regularizes the policy to boost (suppress) winning and losing response likelihood depending on the strength of the preference signal.

---

> ### Author Response · Authors · 2024-11-21
> **Sensitivity of hyperparameters in DRDO**
>
> **Sensitivity of hyperparameters in DRDO training (eq. 6):**
>
> We would also like to clarify that as outlines in section 4 (footnote 4) that DRDO theoretically does *not* necessitate an $\alpha$ parameter in the traditional sense of correcting class imbalance [4] since the true preference probabilities are unknown (likely continuous and therefore cannot be categorized into discrete classes). However, similar to [4], we find that keeping an alpha of 0.1 stabilises training and as such, we keep $\alpha$ as 0.1 in all our experiments (as mentioned in section 4).
>
> For more clarity, we ran an additional experiment to compare the sensitivity of model-specific hyperparameters. Below, keeping $\alpha$ as 0.1 for all DRDO policies, we compute the KL-divergence during training on sampled generations on 40 randomly sampled evaluation prompts in the held-out set of Ultrafeedback with top-p of 0.8 and temperature of 0.7 with various $\gamma$ values in DRDO ($\alpha = 0.1$) and with different KL-$\beta$ values in DPO using the SFT-trained Phi-3-Mini-4K-Instruct model. The expected rewards with the same hyperparameters over these samples are shown in the second table below.
>
> DRDO vs DPO wrt KL divergence over the SFT (reference) model.
> | Step | DRDO ($\gamma = 5$) | DRDO ($\gamma = 2$) | DRDO ($\gamma = 1$) | DPO ($\beta = 0.01$) | DPO ($\beta = 0.1$) |
> |------|----------------------|----------------------|----------------------|-----------------------|----------------------|
> | 100  | 0.63                | 0.44                | 0.82                | 0.64                 | 0.38                |
> | 200  | 0.35                | 1.51                | 1.67                | 0.81                 | 0.39                |
> | 300  | 0.59                | 1.67                | 1.82                | 1.17                 | 0.42                |
> | 400  | 0.64                | 1.63                | 1.71                | 1.35                 | 0.44                |
>
> DRDO vs DPO expected rewards with the OPT 1.3B oracle model
>
> | Model                   | Expected Oracle Reward       |
> |-------------------------|-----------------------------|
> | DPO ($\beta = 0.1$)     | 0.775 ($\pm$ 0.42)         |
> | DPO ($\beta = 0.01$)    | 0.675 ($\pm$ 0.47)         |
> | DRDO ($\gamma = 1$)     | 0.750 ($\pm$ 0.44)         |
> | DRDO ($\gamma = 2$)     | 0.825 ($\pm$ 0.38)         |
> | DRDO ($\gamma = 5$)     | 0.600 ($\pm$ 0.50)         |
>
> These two result tables suggest that while DRDO does not explicitly regularise its policy wrt reference-model based KL regularization, it still outperforms DPO in oracle-assigned expected reward accuracies (win-rates) on sampled generations as long as the $\gamma$ parameter is carefully chosen. In particular, as observed in [1,2], smaller $\beta$ in DPO tends to increase KL-divergence with respect to the baseline SFT model. However, a relatively larger KL divergence in DRDO on average does not necessarily impede preference learning but larger $\gamma$ values tend to degrade expected rewards.
>
> [1] Meng, Y., Xia, M., & Chen, D. (2024). Simpo: Simple preference optimization with a reference-free reward. arXiv preprint arXiv:2405.14734.
>
> [2] Rafailov, R., Sharma, A., Mitchell, E., Manning, C. D., Ermon, S., & Finn, C. (2024). Direct preference optimization: Your language model is secretly a reward model. Advances in Neural Information Processing Systems, 36.
>
> [3] Adam Fisch, Jacob Eisenstein, Vicky Zayats, Alekh Agarwal, Ahmad Beirami, Chirag Nagpal, Pete Shaw, and Jonathan Berant. Robust preference optimization through reward model distillation, 2024. URL https://arxiv.org/abs/2405.19316.
>
> [4] Ross, T. Y., & Dollár, G. K. H. P. (2017, July). Focal loss for dense object detection. In proceedings of the IEEE conference on computer vision and pattern recognition (pp. 2980-2988).
>
> [5] Yang, Rui, et al. "Regularizing Hidden States Enables Learning Generalizable Reward Model for LLMs." arXiv preprint arXiv:2406.10216 (2024).

---

### Author Response · Authors · 2024-11-23

We thank the reviewers for taking the time to closely read our paper and post their feedback. We have responded in depth to each review and we look forward to engaging in the author-reviewer discussion in the coming days.

---

### Author Response · Authors · 2024-11-24

As there are just a few days left on the discussion period, we hope the reviewers have had a chance to read our response, and we look forward to having a productive discussion with you in the remaining days.

---

### Author Response · Authors · 2024-11-26
**Summary of revisions and additional results for DRDO**

We thank the reviewers for their feedback and insightful comments. In particular we are delighted that they found DRDO to be novel (reviewer **G9Er**), very relevant wrt preference alignment in LLMs (reviewer **z2iA**), theoretically and mathematically sound (reviewers **aDcH**, **z2iA**) and robustly evaluated (reviewers **aDcH**, **Pipq**, **z2iA**) with well-designed experiments across datasets. Here we briefly outline the changes made to the manuscript (in red font and including both additional experimental results and presentation fixes) and some crucial points brought up in the reviews. We hope that in the remaining time we will have the opportunity to address any remaining concerns and have a fruitful discussion.

**Main paper**
- **Page 1 (introduction)**: added clarifications on DRDO motivation (in light of existing methods) as suggested by **G9Er**
- **Lemma 1a) (theoretical motivation in sec.3)**: changed lemma 1a statement to reflect correction as suggested by  **G9Er**
- **Non-deterministic preferences (page 4)**: clarified definitions of non-deterministic preferences in typical preference datasets vs flipped-label based noise  (**Pipq**)
- **Oracle/student policy details (sec. 4, page 6)**: clarified reward estimate computations using oracle/student policy including explicit parametrizations including minor change in eq. 5 suggested by **aE9X**

- **Constrastive log-unlikelihood (Fig. 2)**: replotted Fig. 2 with correction on y-axis label (**G9Er**)
- **OOD specification (Page 7)**: added a line clarifying why CNN/Daily Mail prompts constitute an OOD or out-of-domain settings for our experiments (**aDcH**)

**Appendix**
We have slightly rearranged the order in the appendix and added new sections for more clarity.

- **Appendix A.2**: corrected proof of lemma 1a and 1b (**G9Er**)
- **Appendix A.4**: corrected proof of lemma 2 (**G9Er**)
-**Appendix A.5**: added figure for DRDO complete pseudo-code/algorithm  (**Pipq**)
- **Appendix E.1 and Table. 8**: added DRDO component ablations results along with IPO baseline with a larger oracle (GPT-4o) (suggested by **z2iA**, **aE9X** and **Pipq**)
- **Appendix E.2 and Fig. 5**: added DRDO comparison with baselines (DPO and e-DPO) with extent of OOD generalization (**aDcH**)
- **Appendix F**: added clarifications and comment about DRDO’s computational overhead vs baselines (**aE9X**)
- **Appendix G**: added clarifications on DRDO’s motivation vs ensemble/linear interpolation based pluralistic preference learning (**G9Er**)
- **Appendix E.1 and Fig. 8**: added evaluation prompt using a larger oracle (GPT-4o) to resolve concerns regarding capacity and data–sufficiency in oracle  (**Pipq**).
- **Appendix H and Table 9 and 10**: added further results and analysis (KL-divergence and expected reward accuracies over SFT baseline) comparing DRDO’s $\gamma$ sensitivity vs DPO’s $\beta$ hyperparameter as suggested by **z2iA**.

---

### Author Response · Authors · 2024-11-29

Dear reviewers,

As the discussion period has been extended until December 2nd, we would appreciate it if you could review our revisions and additional comments on the discussion board, and please let us know if and how we can address your questions further. We thank you for your reviews and comments to date and are confident that our revisions accordingly have resulted in a stronger paper and would appreciate your further engagement for the remainder of the discussion period.

Many thanks,
The authors.

---

### Meta-Review · Area_Chair_4shf · 2024-12-21

**Metareview:**

This paper provides a new loss for simultaneous reward distillation and preference learning. The main paper is well-written and the experimental evidence is strong further motivating the proposal. However, the theoretical derivations are hard to follow, containing steps that are not verifiable. In particular, Reviewer G9Er worked through several of the derivations with the authors and helped them fix issues. However, unfortunately there are still remaining concerns (verified by the AC) in the proofs (as per the last comment of Reviewer G9Er). Given the issues in theoretical correctness, the paper unfortunately has to be recommended for rejection at this point. We hope that the authors can revise the paper using this feedback for a future venue.

**Additional Comments On Reviewer Discussion:**

Reviewer G9Er has identified flaws in theoretical claims/derivations that are blockers to publication (and I verified the concerns to be valid), so no further discussion is needed.

---

### Decision · Program_Chairs · 2025-01-22

Reject